# Monitoring and Predicting Channel Morphology of the Tongtian River, Headwater of the Yangtze River Using Landsat Images and Lightweight Neural Network

Bin Deng [1,2,3], Kai Xiong [1], Zhiyong Huang [1,2,3,*], Changbo Jiang [1,2,3], Jiang Liu [1], Wei Luo [1] and Yifei Xiang [4]

1. School of Hydraulic and Environmental Engineering, Changsha University of Science & Technology, Changsha 410114, China; dengbin07@csust.edu.cn (B.D.); cdpanda2500@stu.csust.edu.cn (K.X.); jiangchb@csust.edu.cn (C.J.); liujiang@stu.csust.edu.cn (J.L.); luowei126@csust.edu.cn (W.L.)
2. Key Laboratory of Dongting Lake Aquatic Eco-Environmental Control and Restoration of Hunan Province, Changsha 410114, China
3. Key Laboratory of Water-Sediment Sciences and Water Disaster Prevention of Hunan Province, Changsha 410114, China
4. College of Artificial Intelligence, Guangxi Minzu University, Nanning 530006, China; 2020210812001026@stu.gxun.edu.cn
* Correspondence: huangzy9084@csust.edu.cn

**Abstract:** The Tongtian River is the source of the Yangtze River and is a national key ecological reserve in China. Monitoring and predicting the changes and mechanisms of the Tongtian River channel morphology are beneficial to protecting the "Asian Water Tower". This study aims to quantitatively monitor and predict the accretion and erosion area of the Tongtian River channel morphology during the past 30 years (1990–2020). Firstly, the water bodies of the Tongtian River were extracted and the accretion and erosion areas were quantified using 1108 Landsat images based on the combined method of three water-body indices and a threshold, and the surface-water dataset provided by the European Commission Joint Research Centre. Secondly, an intelligent lightweight neural-network model was constructed to predict and analyze the accretion and erosion area of the Tongtian River. Results indicate that the Tongtian River experienced apparent accretion and erosion with a total area of 98.3 and 94.9 km$^2$, respectively, during 1990–2020. The braided (meandering) reaches at the upper (lower) Tongtian River exhibit an overall trend of accretion (erosion). The Tongtian River channel morphology was determined by the synergistic effect of sediment-transport velocity and streamflow. The lightweight neural network well-reproduced the complex nonlinear processes in the river-channel morphology with a final prediction error of 0.0048 km$^2$ for the training session and 4.6 km$^2$ for the test session. Results in this study provide more effective, reasonable, and scientific decision-making aids for monitoring, protecting, understanding, and mining the evolution characteristics of rivers, especially the complex change processes of braided river channels in alpine regions and developing countries.

**Keywords:** water-body indices; lightweight neural network; river-channel morphology; accretion and erosion area; the Tongtian River

## 1. Introduction

The Three-River-Source Region (TRSR, i.e., the source of the Yangtze River, Yellow River, and Lancang River) is located in the center of the Tibet Plateau, China. Despite the vulnerable environment, this region has very important ecological functions for the sustainable development of the middle and lower reaches of the TRSR. The Tongtian River is located at the source of the Yangtze River with more than 200 tributaries. The Tongtian River is characterized by braided morphology, and it is the main part of the National Park of the TRSR. Accretion and erosion are key processes that influence river stability and determine the characteristics of river-channel evolution. Studying the accretion and

erosion processes of river-channel morphology is essential to accurately interpret the current status of rivers and predict future trends [1–3]. This study aims at investigating the evolution characteristics of the erosion and accretion processes of the Tongtian River channel morphology, which can deepen the understanding of the basin status of the source of the Yangtze River and promote ecological protection and management of the TRSR.

Some scholars have used remote-sensing data to study the morphological changes in the Tongtian River. For instance, Li et al. [4] interpreted Landsat remote-sensing images and figured out that the planar morphology of the local braided reaches of the Tongtian River is positively correlated with the branching intensity, which is strongly related to streamflow. Li et al. [5] extracted water bodies of the two main braided rivers (the Tongtian River and Ulan Moron) located at the source of the Yangtze River using the Modified Normalized Difference Water Index (*MNDWI*) and found similar morphological changes in these two rivers. Li et al. [6] used remote-sensing images and hydrological data from 2006 to 2017, combined with the *MNDWI* to extract the water body of the local braided reaches of the Tongtian River, and found that the cutting of the group sandbars in the braided channel was due to the floodplain erosion during the flooding period and the water-level difference between the secondary branches and the main branch. Jiang et al. [7] used Landsat images and hydrological data from 2002 to 2015 to explore the morphological changes of the two typical braided channels, the Ulan Moron and Tongtian River, and found that the side bars developed attaching to two banks can be classified into three types, i.e., stable, substable, and active side bars. Previous studies only analyzed the local morphological changes and associated reasons of the Tongtian River and did not focus on the erosion and accretion processes of the Tongtian River channel morphology at the large spatial scale and multitemporal scales. Moreover, previous studies did not involve the prediction of the channel morphology of the Tongtian River.

A crucial step to monitoring the river-channel morphology using remote-sensing images is extracting the water-body extent of the river channel, generally based on the water-body index. Previous studies mainly used four water-body indices, namely *MNDWI*, Normalized Difference Water Index (*NDWI*), Normalized Difference Vegetation Index (*NDVI*), and Enhanced Vegetation Index (*EVI*) to extract water bodies from remote-sensing images. For example, Zhang et al. [8] analyzed the long-term spatiotemporal dynamics of surface water in the Hanjiang River based on the *MNDWI*. Deng et al. [9] analyzed the seasonal changes in reservoir water storage in two basins of the upper Yellow River based on the *NDWI*. Gao et al. [10] discussed the relationship between the main branch changes and flow in the local braided reaches of the Lancang River based on the *NDWI*. Considering different performances of different water-body indices; some researchers extracted the water body by selecting the optimal index among several water-body indices. For instance, Talukdar and Pal [11] used the *NDWI* and *MNDWI* to obtain a wetland map of the study area during the same period and selected the optimal map with higher accuracy. Worden and Beurs [12] tested four water-body indices (*NDWI*, *MNDWI*, *EVI*, *NDVI*) using Landsat images, and finally chose the *MNDWI* for surface-water detection in the Caucasus. Only a few studies have attempted to integrate multiple water-body indices to improve the accuracy of water-body extraction. For instance, Yang and Ke [13] extracted the water body of a lake by jointly using four water-body indices (i.e., *NDWI*, *MNDWI*, Revised normalized difference water index, and spectrum-photometric method) and the Bayesian decision theory. Huang et al. [14] analyzed the dynamic changes of the surface-water extent of the Irtysh River Basin by taking the following criteria to extract and record the water body: *MNDWI* > *EVI* or *MNDWI* > *NDVI* and *EVI* < 0.1.

In addition to using the water-body indices to extract water bodies from remote-sensing images, some scholars or communities have investigated and released relevant surface-water datasets with free access to the public. For instance, the Joint Research Center (JRC) [15] jointly used expert systems, visual analytics, and evidential reasoning methods to extract the global surface-water bodies using millions of Landsat images since 1984 with a false rate below 1% and a missing rate below 5%. This dataset has been used by many

scholars. For instance, Walker et al. [16] used the JRC global surface-water dataset (hereafter defined as "the JRC dataset") to obtain the time series of surface-water extent in the Central Valley of California. Li et al. [17] obtained bathymetric data for the dynamic areas of global reservoirs by combining the altimetric elevation and JRC dataset. Although the JRC dataset is freely available, it is still unclear whether its accuracy is better than the water bodies extracted using a single water-body index or the joint use of multiple water-body indices. Therefore, it is a promising topic to compare the performance of the different water-body indices (single- or joint-use) and the JRC dataset for monitoring accretion and erosion area change of river-channel morphology.

The development of artificial-intelligence technology has driven some scholars to investigate the evolution of river-channel morphology based on machine learning. For instance, Abid et al. [18] identified water bodies using an unsupervised curriculum-learning method based on the convolutional neural network, which overcomes the challenges faced by remote-sensing images. Hosseiny [19] proposed a novel deep-learning framework for automatic identification of river geometry and flooding extent and prediction of river-water depth. Aziz et al. [20] used the unsupervised machine-learning algorithms, such as the k-means, clustering large application, and hierarchical agglomerative clustering for predicting river-sediment adaptation. Ahmed et al. [21] used the same methods as used by Aziz et al. [20] as well as the self-organizing tree algorithm to perform supervised classification for satellite images and predict future river-sedimentation areas. However, there are some limitations for these machine-learning methods. For instance, the model structure of deep learning is much more complex than that of the data-mining algorithm, and the image-processing processes require a heavy computational burden. Compared with the neural-network model, the unsupervised-learning model cannot extract the nonlinear relationship among variables well and cannot mine the relationship among variables from a higher dimension. Moreover, previous studies did not deeply analyze the change process and law of the river-channel morphology and associated reasons both quantitatively and qualitatively from the perspective of data mining.

Compared with the above-mentioned machine-learning methods, there is another simpler and more efficient method, namely the lightweight neural network. This method does not need the pooling layer during image processing and greatly reduces the number of hidden layers and neurons, and only requires a few layers of neural networks to achieve complete training of complex nonlinear data. Moreover, the lightweight neural network can improve the training speed by saving a lot of time needed for parameter updates during the training process. Further, this method can be used to deeply excavate the relationship between hydrological conditions and channel morphology with high precision, so as to qualitatively and quantitatively analyze their inherent implicit relationship and predict the channel morphology changes. The lightweight neural network has been applied to water-body extraction from remote-sensing images and image change detection [22–24], but has not been applied to predicting the accretion and erosion area change of river-channel morphology.

This study focused on both quantitatively and qualitatively monitoring, analyses, and prediction of the accretion and erosion area changes of the Tongtian River channel morphology at a large spatial scale and multitemporal scales using Landsat images (1990–2020) and the lightweight neural-network model. The optimal method was selected for water-body extraction of different reaches of the Tongtian River by comparing three methods, i.e., a single water-body index (e.g., *NDVI* or *EVI*), the combined method of the three water-body indices (i.e., *MNDWI*, *NDVI*, and *EVI*) and a threshold (hereafter defined as "the combined method"), and the JRC dataset. Compared with the image-processing tools such as ArcGIS, ENVI, and Matlab, which have a slow speed for calculating water-body indices, the Google Earth Engine (GEE) is much more efficient because the cloud computing platform integrates a large number of remote-sensing datasets including Landsat and it possesses millions of servers around the world with the most advanced cloud computing capability and storage [25–27]. Therefore, this paper intends to use the GEE platform for

batch extraction of water bodies of the Tongtian River. This study intends to answer the following two key questions: (1) What are the spatiotemporal characteristics, trends, and associated influencing factors of the accretion and erosion-area change of the Tongtian River channel morphology? (2) Can the lightweight neural-network model well predict the accretion and erosion-area change of the Tongtian River channel morphology, and how is the prediction accuracy?

## 2. Materials and Methods

### 2.1. Overview of the Study Area

The area of lakes, swamps, and glaciers of the TRSR accounts for more than 89% of the total area of the TRSR. The altitude of the TRSR ranges from 3531 to 6575 m. The annual average temperature in this region is 4.8 °C and the annual precipitation is 397.8 mm [28]. The source area of the Tongtian River consists of the Ulan Moron River in the west, Chuma'er River in the north, and Dangqu River in the south (Figure 1). The Tongtian River is the largest mainstream in the source area of the Yangtze River, and changes in its runoff and baseflow reflect the hydrology and hydrogeological characteristics of the entire source area. The glacier reserves in the Tongtian River Basin are 149.6 km$^3$. The main stream of the Tongtian River is 1174 km long, and the total area of this basin is about 140,000 km$^2$.

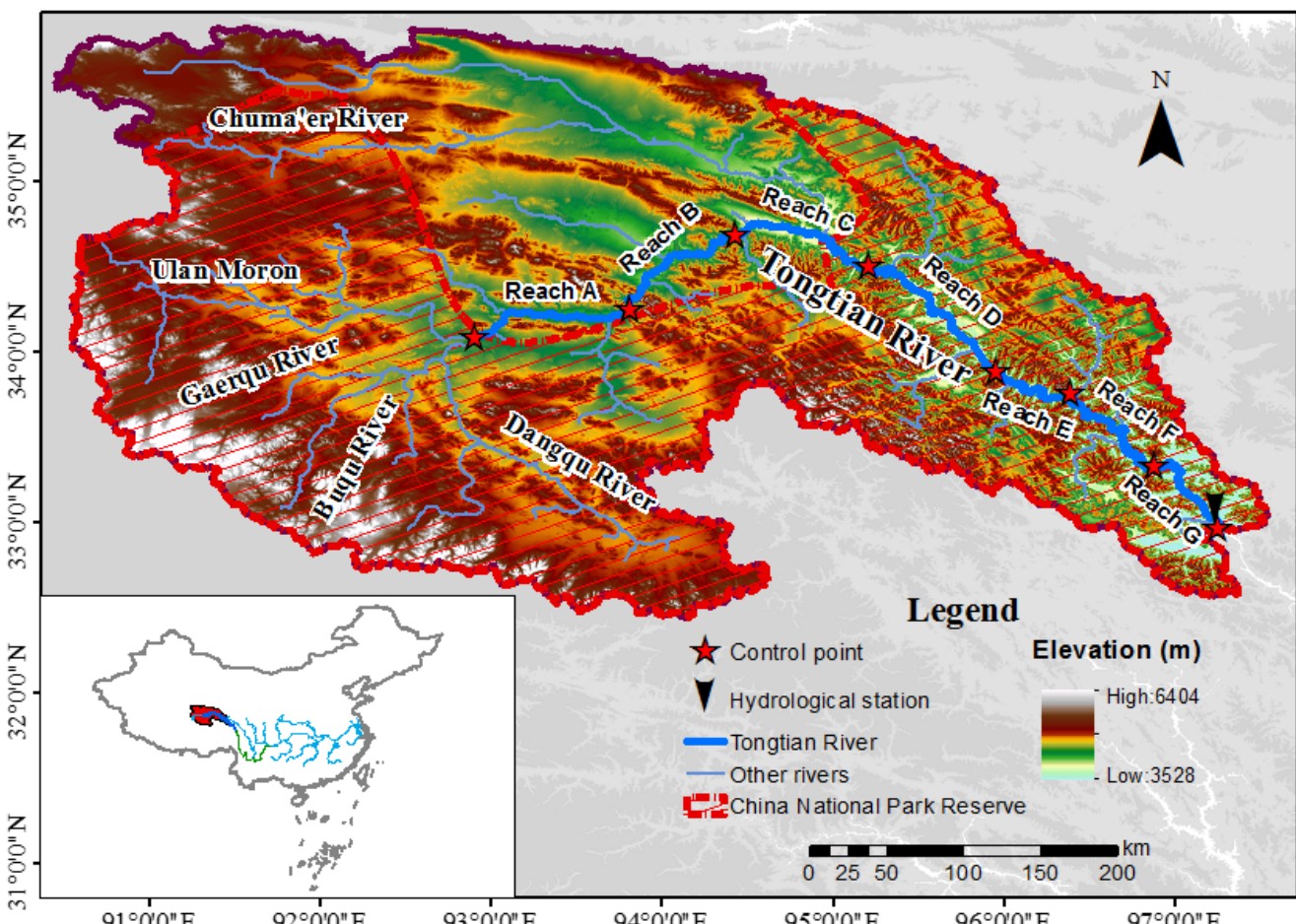

**Figure 1.** Overview of the study area, the Tongtian River Basin at the source of the Yangtze River. The red pentagrams represent the eight control points selected in this study. The black inverted triangle at the outlet of the lower Tongtian River is Zhimenda Hydrological Station. The inserted map on the lower-left corner shows the location of the Tongtian River in China and the major rivers of the Yangtze River.



The TRSR covers an area of 302,500 km$^2$, and the source of the Yangtze River accounts for 44% of the TRSR area. A quarter of the total water resources in the Yangtze River come from TRSR, which is also an important water source for China and even the rest of Asia. To curb the deterioration of ecological conditions of the TRSR, increase the vegetation coverage, and improve the biodiversity, in 2005, the State Council of China approved the "Overall Plan for Ecological Protection and Construction of the Nature Reserves of the TRSR in Qinghai". The first phase of the project was implemented from 2005 to 2013. The total investment is RMB 7.5 billion. The implementation area of the project is 150,000 km$^2$, accounting for 49.6% of the area of TRSR. In 2008, the State Council of China put forward the requirements of "Starting the preliminary research work of the second phase of the TRSR project in a timely manner" and "Establishing a comprehensive experimental area for the national ecological protection of the TRSR". The planning period is from 2013 to 2020. The total investment increased to 16 billion RMB, and the total planned construction area is 395,000 km$^2$, covering the whole area of TRSR, and the planned area accounts for 54.6% of the total area of Qinghai Province.

### 2.2. Landsat Images

The whole study started from the confluence of the Ulan Moron River and Dangqu River; that is, the starting point of the Tongtian River, until the Tongtian River merges into the Jinsha River, covering 7 Landsat scenarios at different paths and rows (Table S1). All steps of water-body extraction from Landsat images (also JRC datasets) were operated on the GEE platform. The same first step was to map out the research area of interest: the entire research area started at 92.907°E, 34.086°N, and ended at 97.248°E, 32.963°N, forming a map of the entire Tongtian River Basin. A total of 113 plotted points were drawn for the study area. The line segment formed by two plotted points is parallel to the edge of the Tongtian River. On the premise that the Tongtian River can be completely mapped within the drawn study area, other tiny rivers, lakes, and their tributaries should be avoided as far as possible in the drawn study area.

For Landsat images, taking into account the availability of cloud-free scenes and the acquisition period of Landsat, a total of 1108 Landsat images were acquired between 1990 and 2020 (Table S2). The default resolution of Landsat images is 30 m, which will greatly affect the accuracy of machine learning because river-channel morphology research requires satellite images with high spatial resolution and low cloud cover. If it is only used for use in ENVI, ArcGIS, and other software to improve the accuracy of the mask, Landsat 4/5 and the defective Landsat 7 cannot meet the research needs. Therefore, this study repaired and improved the accuracy of Landsat data through GEE, and achieved satisfactory observation results. All scenes are primary datasets of Landsat Collection 1, which can be obtained through the database provided by the USGS website on the GEE platform. All Landsat images are from May to October each year (non-icing period).

### 2.3. Extracting Water Bodies from Satellite-Based Databases

Figure 2 shows the flow chart of the methods and techniques used in this study. While about 1100 Landsat images were obtained covering the Tongtian River, many of them were heavily influenced by clouds or cloud shadows, snow cover, or snowmelt. All images were processed online using the GEE platform. During the generation of the dataset, preprocessing such as radiometric calibration, atmospheric correction, and the geometric correction was performed. To ensure the visibility and analyzability of remote-sensing images, the cloud-cover threshold was set to 5% in this study. All the Collection 1 data in the Landsat series make it easier to identify suitable scenarios for pixel-level analysis of time series. The T1 class data in the Level-1 dataset have the highest available data quality, Level 1 accuracy, terrain correction, and excellent radiation measurement values. The most important thing is that the radial root-mean-square error between this type of data and the real data is less than 12 m [29].

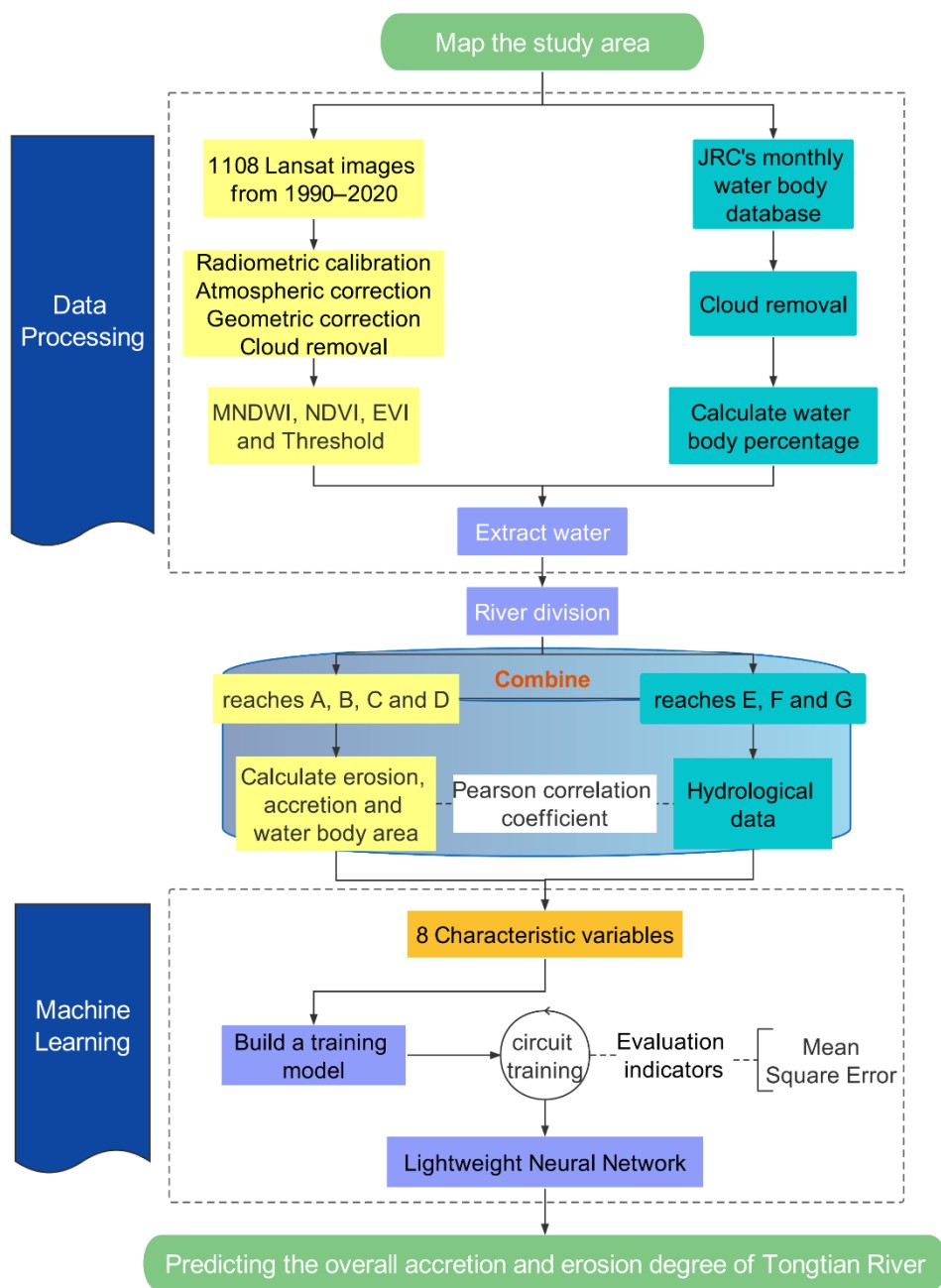

**Figure 2.** Flowchart of methods and techniques in this study.

This study jointly used three water-body indices (the *MNDWI* [8], *NDVI* [30], and *EVI* [31]) to extract water-body information from Landsat_Collection 1_T1_SR remote sensing imagery. Although some studies have shown that the water-body extraction performance of *MNDWI* is higher than that of *NDWI* and *EVI*, *MNDWI* is not 100% effective, especially in alpine regions with snow cover. Therefore, to ensure that the water body of Tongtian River can be extracted as much as possible, a joint calculation of multiple indicators (i.e., the combined method) was performed. Their calculation priority is *MNDWI* > *EVI&MNDWI* > 0.1 > *EVI&NDVI*. Here, the value 0.1 refers to the threshold following previous studies [14,32,33]. That means when *MNDWI* cannot extract the effective water body, it can be extracted through the joint calculation of *EVI* and *MNDWI*. Similarly, when the combined calculation of *EVI* and *MNDWI* still cannot effectively extract the water

body, the remaining indicators can be used to extract the water body to form a complete river-water-body map. The three water-body indices can be calculated as follows:

$$MNDWI = ((Green) - (MIR))/((Green) + (MIR)) \tag{1}$$

$$NDVI = (NIR - R)/(NIR + R) \tag{2}$$

$$EVI = 2.5 \times ((NIR - R)/(NIR + 6 \times R - 7.5 \times Blue + 1)) \tag{3}$$

where *Green*, *Blue*, *NIR*, *R*, and *MIR* are the green, blue, near-infrared reflectance, infrared reflectance, and mid-infrared reflectance of Landsat image, respectively. To reduce the impact of fringe noise in Landsat-7 images and improve the accuracy of water-body extraction from all remote-sensing images, the water-body images extracted from May to October each year through the combined method are synthesized using the reducer module in GEE to form an inter-annual water-body image. The operation procedure is as follows: Sum up the pixel values in the same grid in all water images; when all pixels in the grid are summed, a composite image can be obtained.

Located at the TRSR, the Tongtian River is covered by a large amount of surface snow [34,35], and there is a large amount of surface runoff generated by snow melting in summer. The river width increases due to the increase in river flow in summer. Therefore, to ensure that the Tongtian River channel can be completely extracted, the surface runoff should be excluded in the mapped study area as far as possible. To extract the water bodies more accurately, a buffer zone of 50 m is set up along the edge of the river channel to cover all the water bodies of the river channel. The buffer zone is set to be larger than the resolution (30 m) of the Landsat images and less than the distance of 2 pixels (60 m). This buffer zone is reasonable because a larger buffer zone will increase the computation burden for water-body identification and classification, and a smaller buffer zone with only one pixel (30 m) is unable to encompass the water body with a distance further than 30 m to the edge of the river channel. Furthermore, with the 50 m buffer zone, the channel edge (water-body edge) can be completely extracted even under the condition of a sudden increase in flow rate.

Because the research period of this study is from May to October every year, the JRC's annual water body is not used; instead, the JRC's monthly water-body database is selected. Therefore, the monthly water bodies from May to October need to be merged to form interannual water bodies. The specific steps for processing the JRC dataset are: (1) Obtaining remote-sensing images of the study area: Control points on the GEE platform are added to form a closed study area covering the entire Tongtian River. At the same time, a buffer zone of 50 m is set up along the edge of the river channel to cover all the water bodies of the river channel. (2) Acquisition of water-body information: A feature attribute is added to each pixel of the monthly water-body remote-sensing image, which is used to record the number of water bodies at each pixel point in the 6 scene images from May to October. If the pixel has data, it is recorded as 1; otherwise, it is recorded as 0. Then, a new feature attribute is added on this basis to perform statistics on the image data with water bodies. According to the new feature attribute, if the number of times of the water-body information in a single pixel of the 6 scene images is ≥3 times, that single pixel is recorded as the Tongtian River. (3) Through the reducer module in GEE, the monthly water body images from May to October of each year are synthesized to form an annual water image. (4) The multitemporal water body image of the Tongtian River is exported and saved as a TIF file.

### 2.4. Comparing Different Water-Body Extraction Methods for Different River Reaches

The eight control points on the Tongtian River Basin (Figure 1 and Table 1) divide the entire reach of the Tongtian River into 7 reaches. According to the plane morphological characteristics, we divided the river channel into braided reaches (A–D) and meandering reaches (E–G). Among the braided reaches, reaches A–C show a high intensity of morphological change and are classified as main braided reaches, while reach D shows a

weak intensity of morphological change, and is classified as minor braided reach (Table 1). Among the meandering reaches, reaches E and F show obvious morphological characteristics and are classified as the main meandering reach, while reach G shows relatively insignificant morphological characteristics and is classified as a minor meandering reach (Table 1).

**Table 1.** Control points of the Tongtian River.

| Control Point | Longitude | Latitude | Channel Type | |
|---|---|---|---|---|
| Starting point | 92.907 | 34.086 | main braided reach (reach A) | |
| 1 | 93.807 | 34.246 | | main braided reach (reach B) |
| 2 | 94.426 | 34.682 | main braided reach (reach C) | |
| 3 | 95.206 | 34.500 | | minor braided reach (reach D) |
| 4 | 95.950 | 33.883 | main meandering reach (reach E) | |
| 5 | 96.387 | 33.755 | | main meandering reach (reach F) |
| 6 | 96.879 | 33.328 | minor meandering reach (reach G) | |
| End point | 97.248 | 32.963 | | |

Figure 3 shows the water bodies for braided reach A and meandering reach G (with enlarged views) extracted from Landsat images in 2020 using the combined method and JRC dataset. By observing the enlarged area of reach A (Figure 3a,b), it can be found that the water body extracted from the JRC dataset is incomplete with braided channel faults, low connectivity, and blurred boundaries relative to the water body extracted from the combined method. In reach G, the water body extracted by the JRC dataset is better, because the channel is smoother and more connected without obvious channel discontinuation (Figure 3c,d). Overall, the water body extracted from the Landsat image by the combined method (the JRC dataset) can better restore the real situation of the braided reaches (meandering reaches). To achieve the best extraction of water bodies of the Tongtian River, this study uses different methods for different river reaches. Specifically, braided reaches A–D adopt the combined method, and meandering reaches E–G adopt the JRC dataset. In addition to comparing the combined method and the JRC dataset, this study further compared the water-body extraction results for the braided reaches based on the single water-body index (e.g., *NDVI* and *EVI*). The relevant result analysis is presented in Section 3.1.

*2.5. Quantification of Accretion, Erosion, and Unchanged Areas during Different Periods*

This study divided the entire study period (1990–2020) into three episodes following previous studies [36,37], namely 1990–2000, 2000–2010, and 2010–2020. This study also focused on the entire period 1990–2020. For each period, the area of erosion, accretion, and the unchanged were detected and quantified using the following methods. Firstly, the data of two research periods (e.g., 1990 and 2000) are crossed; that is, the water-body images are merged to obtain the intersecting part (unchanged water body) of the two remote-sensing images. Secondly, multiyear water-body images (e.g., 1990 and 2000) were superimposed, and the intersecting part of the water in these two years is calculated. The water bodies that intersected in 1990 and 2000 were removed from the body of water in 1990, and the remainder was recorded as accretion. Finally, the water body that intersected in 1990 and 2000 was removed from the body of water in 2000, and the remainder was recorded as erosion. The accretion, erosion, and unchanged areas are quantified using shape files of the extracted water-body information in ArcGIS 10.8.0.

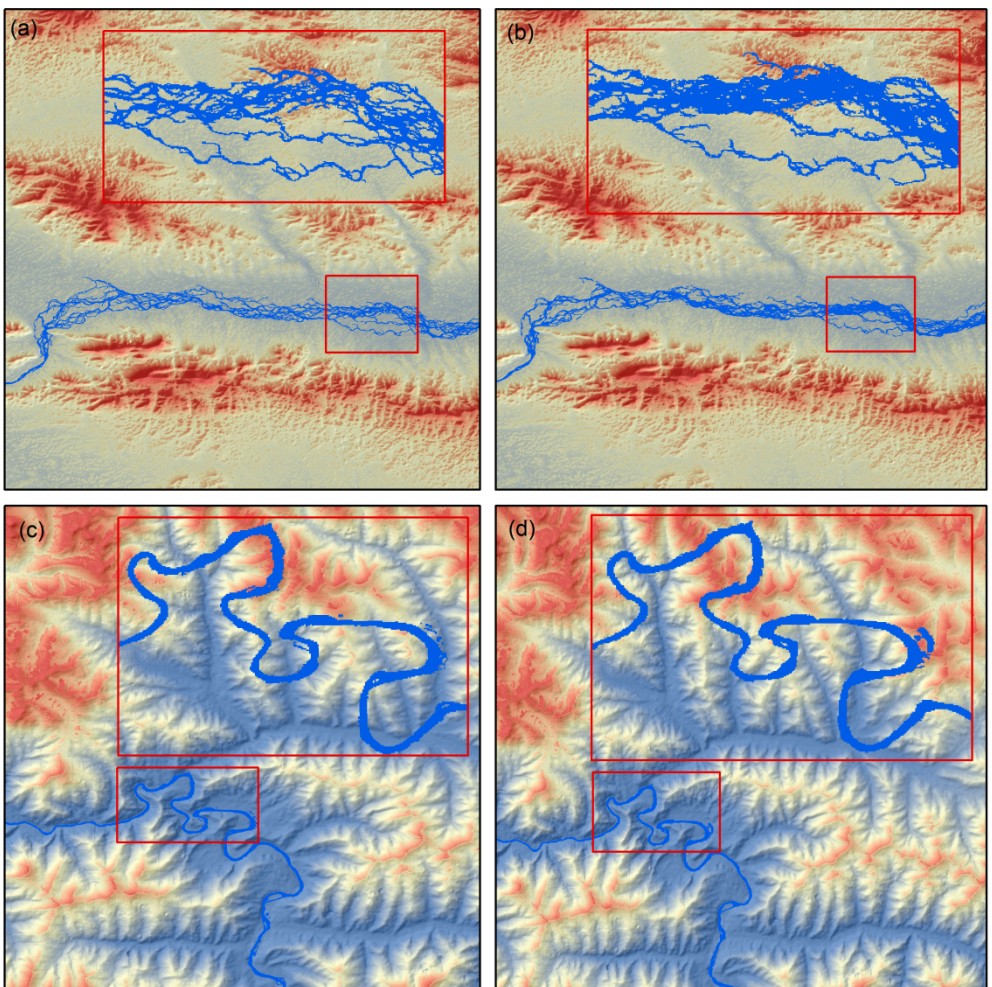

**Figure 3.** Comparison of water bodies in 2020 extracted for reach A (**a,b**) and reach G (**c,d**) of the Tongtian River using the combined method (**a,c**) and the JRC dataset (**b,d**). An inserted image with a red rectangle is used to show the enlarged view of a local reach with a smaller red rectangle.

### 2.6. Statistical Analysis

If the distribution of each feature variable in the data has a strong similarity or is similar after data transformation, then such a data set can enable the machine-learning model to better approach the upper limit of performance. In addition to the quality of the dataset, the correct choice of input parameters and predictive performance of modeling also determines the optimal value for each model and has a significant impact on the predictive performance of the model [38]. Therefore, in this study, according to the large differences in the hydrological data in different dimensions and value ranges, Pearson's correlation coefficient was selected as an index to measure the correlation strength and degree of dependence between all variables. The results are used to test the characteristics of the data; that is, the degree of concentration, dispersion, and distribution shape of the data. The definition of Pearson's correlation coefficient is shown in Equation (4).

$$r(X, Y) = \frac{E[(X - \mu_X)(Y - \mu_Y)]}{\sigma_X \sigma_Y} = \frac{E[(X - \mu_X)(Y - \mu_Y)]}{\sqrt{\sum_{i=1}^{n}(X_i - \mu_X)^2}\sqrt{\sum_{i=1}^{n}(Y_i - \mu_Y)^2}} \tag{4}$$

where $r(X, Y)$ ranges from $-1$ to $1$ (0 means no correlation. A negative value is a negative correlation, and a positive value is a positive correlation). $\sigma_X$ is the standard deviation of $X$. $\sigma_Y$ is the standard deviation of $Y$. $\mu_X$ is the expectation of $X$, and $\mu_Y$ is the expectation of $Y$.

Therefore, Pearson's correlation coefficient can be viewed as the quotient of the covariance and standard deviation between two variables.

*2.7. The Lightweight Intelligent Predictive Model*

Tools for developing the lightweight neural-network model are implemented via Python (v. 3.6.7) using mainstream deep-learning frameworks with a wide range of data science modules, namely TensorFlow (v.1.15.0) and NumPy (v. 1.19.2), pandas (v. 1.1.5), scikit-learn (v. 0.20.1).

The lightweight neural-network model uses 30 years of eigenvectors (i.e., annual averages of sediment transport rate, flow rate, and sediment concentration, and the annual total water-body area) to determine the overall accretion–erosion area change (i.e., the difference between accretion area and erosion area) in the Tongtian River Basin. It is assumed to be a nonlinear regression problem. For the work based on similar predictions found in the literature, the machine-learning model can be made to ignore the underlying physical processes [20,39]. Machine-learning models are more accurate than traditional statistical models and they can be developed and applied faster with little input [40]. The lightweight neural-network model used (1) 30-year annual average hydrological data of sediment transport rate, streamflow, and sediment content obtained from Zhimenda Hydrological Station (1990–2013) (97°14′18″E, 33°00′46″N, Figure 1) managed by the Qinghai Water Conservancy Department and the Bulletin of China River Sediment (2014–2020) provided by the Ministry of Water Resources of China; and (2) total water-body area, the accretion area, erosion area, and unchanged area obtained through the above-mentioned remote-sensing image-processing procedures. Using the data from these two sources, a lightweight neural-network model was built and trained to predict the extent of erosion and accretion in the Tongtian River Basin. The lightweight neural-network model randomly selects the training set and the test set according to the ratio of 7:3 [41–43].

This study refers to related studies of the same order of magnitude for the machine-learning model-building process [41–45]. Given the amount of hydrological data in this study, on the premise of making full use of the implicit function relationship between each input hydrological feature vector and the alluvial area of the Tongtian River, the number of hidden layers of the neural network is set as two to ensure the generalization and robustness of the model. Other parameters to set are epochs, learning rate, and batch. The number and proportion of neurons in different two-layer hidden layers, two activation functions for regression problems, and different optimizers are compared to obtain the best lightweight neural-network model. The model structure is shown in Figure 4. The lightweight neural network uses "Adam" as the optimizer, "Relu" as the activation function ("Adam" and "Relu" are embedded functions in the TensorFlow module in Python), and the evaluation index is mean-square error (MSE) [46], so as to obtain the optimal parameters and network structure that can balance prediction accuracy and training-cost set up.

$$MSE = \frac{1}{N} \sum_{t=1}^{N} (observed_t - predicted_t)^2 \tag{5}$$

Before the model training starts, data preprocessing is also required. Due to the eigenvectors in this study not being in the same order of magnitude as the area of erosion and accretion, the characteristics of water area and areas of erosion and accretion are scaled, and their dimensions are unified into $km^2$. The variables are in the same order of magnitude, which can improve the predictive ability and training ability of the model.

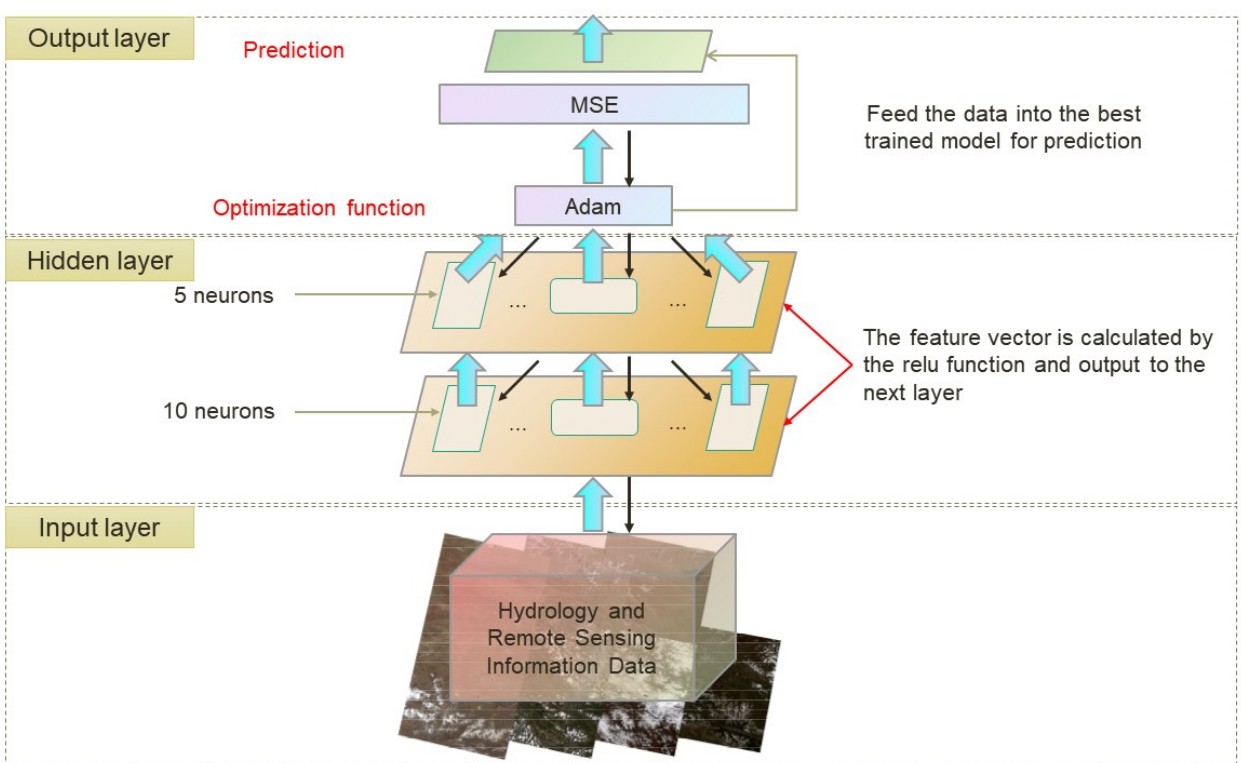

**Figure 4.** Structure diagram of the lightweight intelligent prediction model.

## 3. Results

### 3.1. Accuracy Assessment of Different Water-Body-Extraction Methods for the Braided Reaches

Although Section 2.4 has indicated that the combined method performs better for water-body extraction in the braided reaches than the JRC dataset, it is still unclear whether this method also outperforms the single water-body index method (e.g., the *NDVI* and *EVI*). This section further assesses the accuracy of different methods for water-body extraction of the braided reaches, taking the Landsat images of braided reach A in 1990 as an example for water-body extraction. The water-body-extraction results are shown in Figure 5. As seen, the water body extracted from the JRC dataset (Figure 5b) further shows poorer connectivity than that from the combined method (Figure 5a,e) as demonstrated in Section 2.4. The single water-body index *NDVI* cannot well identify the water bodies in braided reach A with too many missing water-body extents (Figure 5c,e). In contrast to the *NDVI*, the *EVI* method classified many shoals as water bodies, leading to the overestimation of the water-body area for braided reach A (Figure 5d,e).

The total water-body area of the entire Tongtian River extracted using different methods was also quantitatively estimated. Results indicate that the water-body area extracted using the combined method is 2853.17 km$^2$, which is larger than that extracted using the JRC dataset (1656.66 km$^2$) due to better connectivity of the extracted water bodies. The total water-body area of the entire Tongtian River extracted using the *EVI* (*NDVI*) method was 16,189.55 (1164.13) km$^2$, which was significantly overestimated (underestimated). Although there is no in situ data of the water-body area for the Tongtian River for a better accuracy assessment, the above qualitative and quantitative assessments indicate that water bodies extracted for the braided reaches using the combined method are reliable.

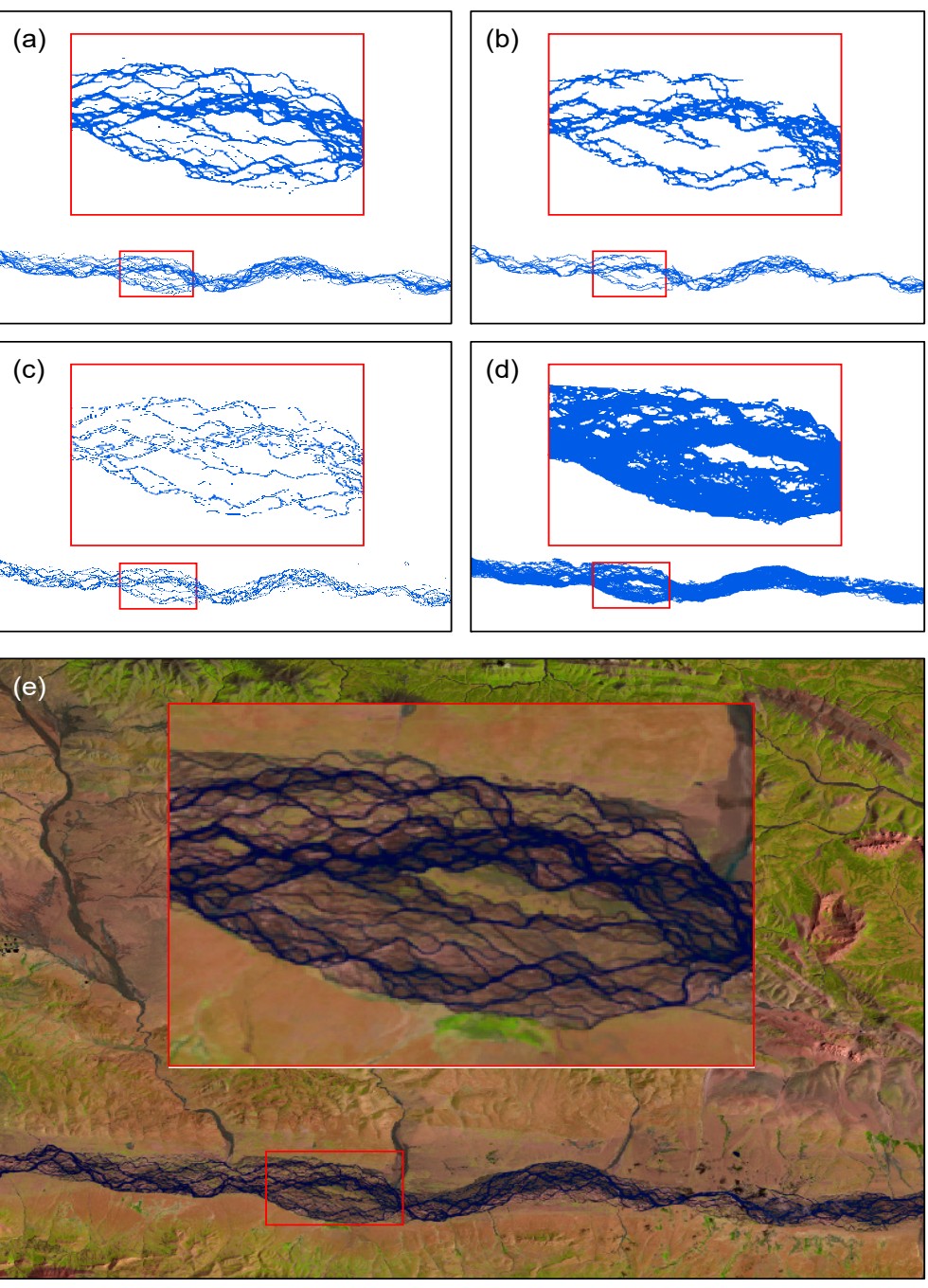

**Figure 5.** Comparison of water bodies for a part of braided reach A extracted from Landsat images in 1990 using different methods: (**a**) the combined method, (**b**) the JRC dataset, (**c**) *NDVI*, (**d**) *EVI*. (**e**) The original Landsat image for a part of reach A on 18 May 1990 with an enlarged view of a local braided reach.

### 3.2. Spatiotemporal Variability of River-Channel Morphology at Different Reaches

Figures 6–8 show the dynamics of historical river erosion and accretion at seven selected reaches during 1990–2020. The results for the three decadal episodes (1990–2000, 2000–2010, and 2010–2020) are shown in Figures S1–S7. Blue represents the unchanged area, red represents erosion, and green represents the accretion of the river course during the study period. Changes in erosion and accretion in the river course during the study period (30 years) were observed from reaches A to G of the Tongtian River. Usually, the river will undergo strong erosion and accretion activities in all braided reaches. Even the river is

cut off, the river distribution is tortuous, and the mainstream of the river changes. These phenomena are usually reflected by the erosion and accretion of the left and right banks. In Figure 6a, the erosion and accretion of river courses on the left and right banks, as well as the river cut-off and the change of the main line are most prominent in reach A in the period 1990–2020, during which the braided shape of the river changed greatly, and the main branch alternatively evolved with the secondary branch. Branches show lateral erosion and swing, and this unstable phenomenon may be attributed to the reoccurrence between erosion and accretion. In general, the main line and the shape of the river course during 2000–2010 (Figure S1b) and 2010–2020 (Figure S1c) changed slightly, showing the law of accretion on the right bank and erosion on the left bank. During 1990–2020 (Figure 6a), after 30 years of high-intensity swings, the river shows a trend of accretion on the right bank in general.

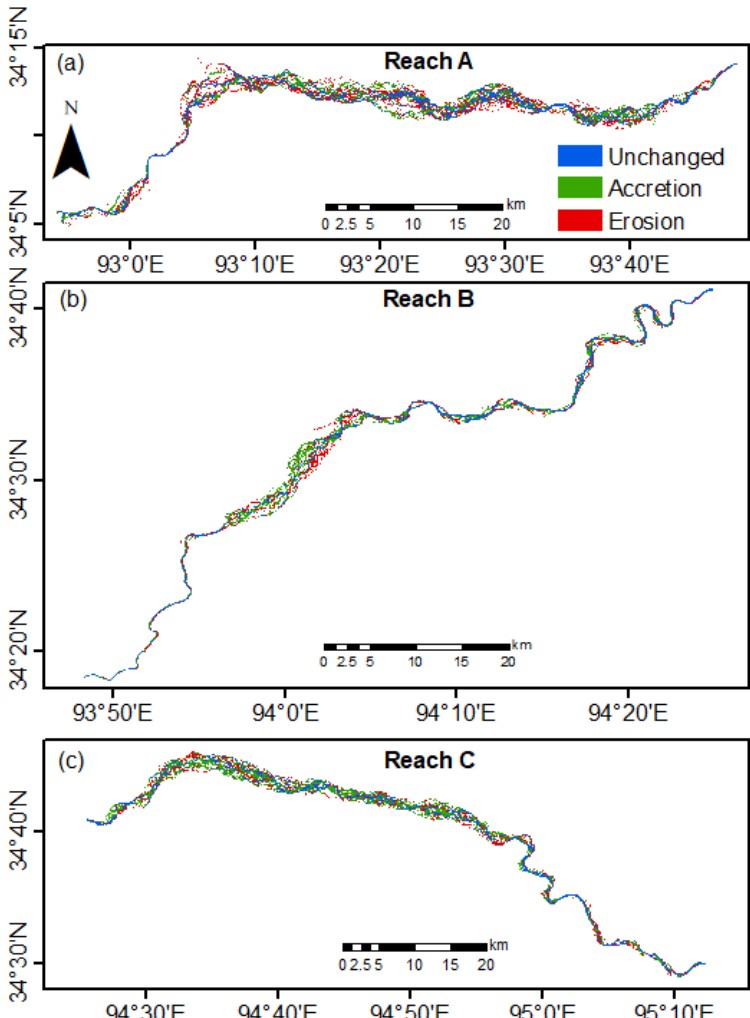

**Figure 6.** Maps showing channel morphology dynamics for the main braided reaches A (**a**), B (**b**), and C (**c**) of the Tongtian River during 1990–2020.

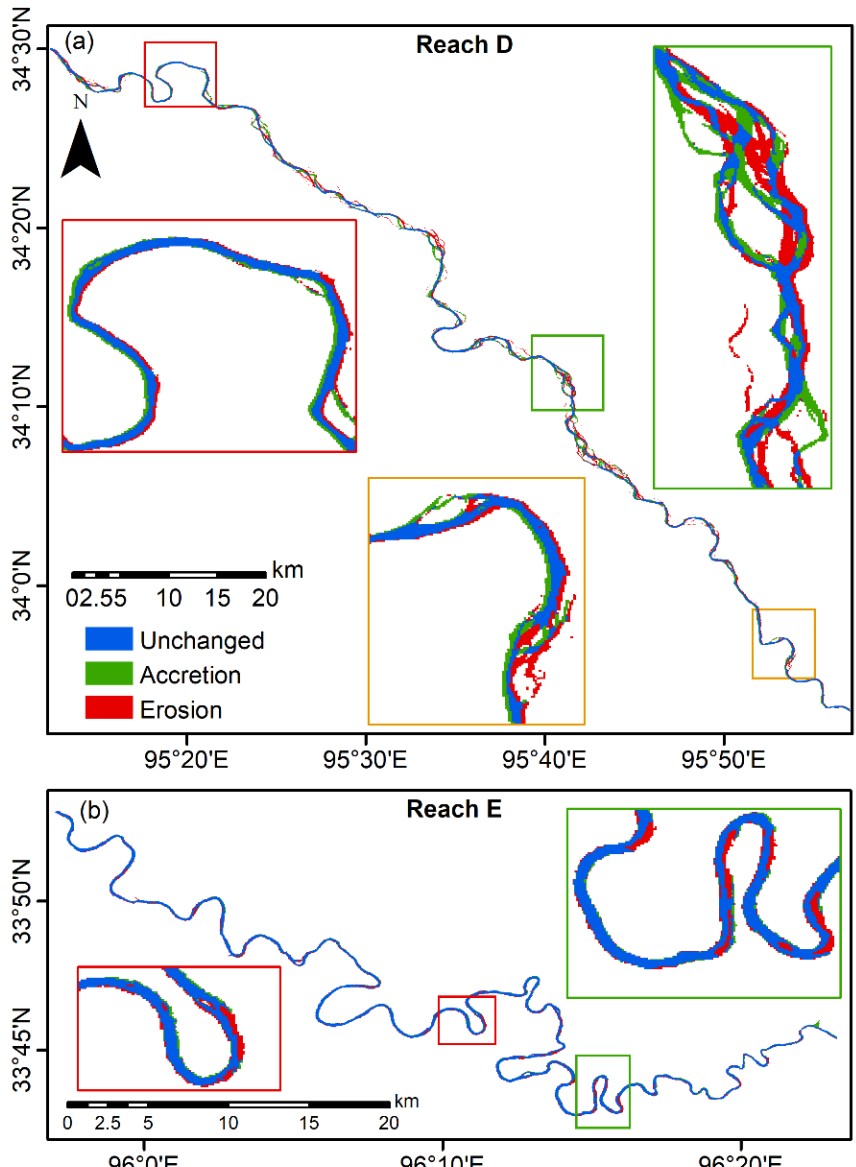

**Figure 7.** The same as Figure 6, but for the minor braided reach D (**a**) and the main meandering reach E (**b**).

Reach B further showed the change of branch direction, cutting of shoals and side shoals by currents in braided channels during 1990–2000 and 2000–2010 (Figure S2a,b). However, it can be observed that the interaction intensity between water and sand conditions and between branches in the channel and sand shoals decreased in 2010–2020 and 1990–2020 (Figures 6b and S2c). During the entire study period (1990–2020), the river course showed a trend of accretion on the left bank in general (Figure 6b).

During 1990–2000 (Figure S3a), the change in reach C mainly occurred in its interior like reach A, which was manifested as the alternating evolution of the main branch and tributary, the cutting of shoals and side shoals by currents, sand shoal erosion and accretion, and change in branch direction. Channel changes in 2000–2010, 2010–2020, and 1990–2020 (Figures 6c and S3b,c) were mainly attributed to the swing of the main river course, and the erosion and accretion on the left and right banks. From 2000 to 2010, the channel morphology of reach C was mainly characterized by accretion on the left bank and partial erosion on the right bank (Figure S3b). From 1990 to 2020, although accretion also generally occurred in reach C, the alternation of its main branches and secondary branches was more

serious than the swing of the river, which was related to the violent alternation of the main tributaries of the river during 2010–2020.

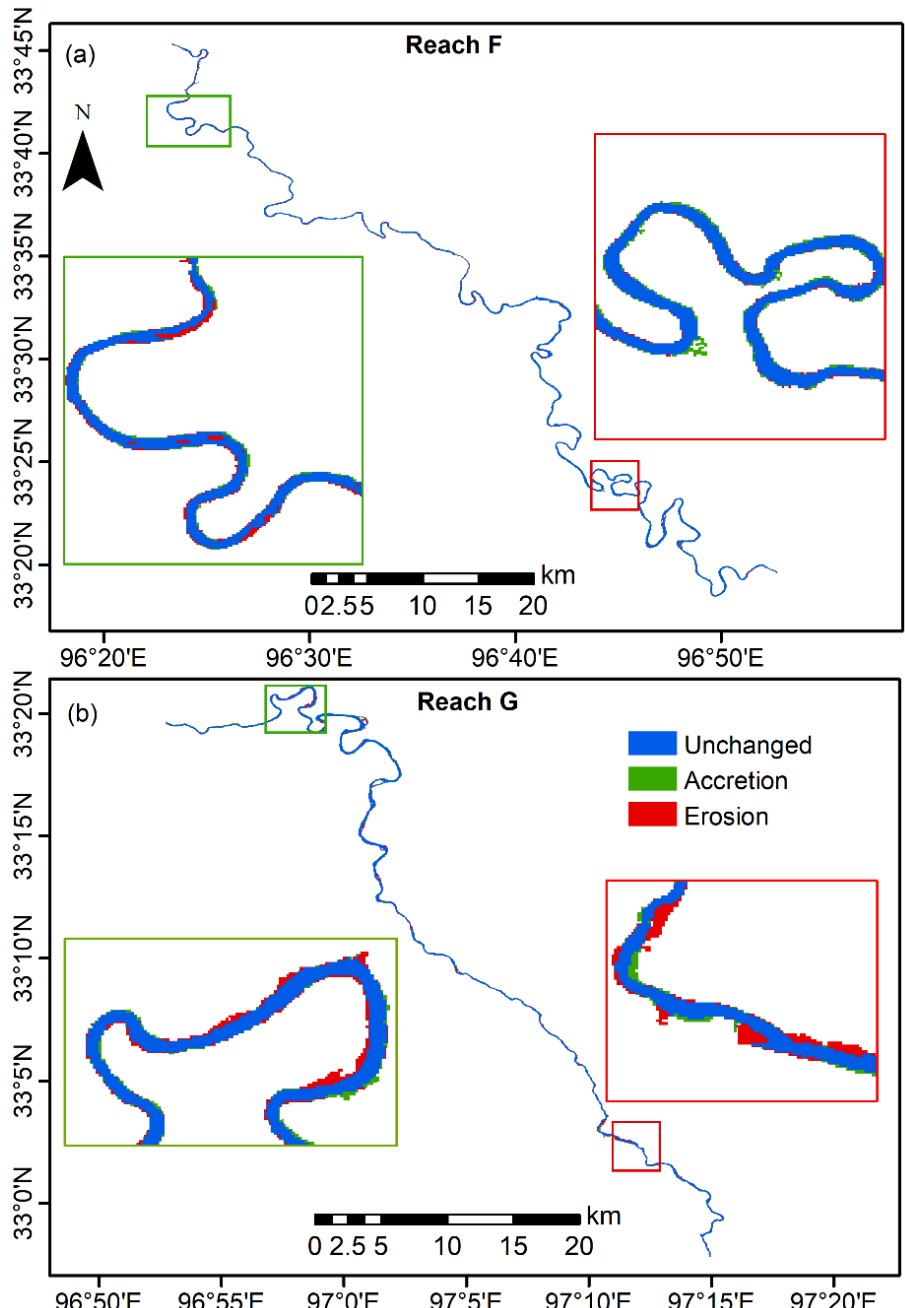

**Figure 8.** The same as Figure 6, but for the minor meandering reaches F (**a**) and G (**b**).

Reach D is a secondary braided channel. During the three episodes of 1990–2000, 2000–2010, and 2010–2020, the major morphological changes lay in the formation or disappearance of new secondary branches (Figure S4a–c). In addition, there were apparent new water bodies formed within the large sandbars as shown in Figure S4b compared to Figure S4a (see the enlarged views with a green rectangle). During 1990–2000 and 2000–2010, the major channel morphological change was accretion, and the intensity of erosion and accretion gradually decreased (Figure S4a,b). From 1990 to 2020 (Figure 7a), the intensity of morphological changes in the river course increased significantly, and the left and right banks had an alternation phenomenon between erosion and accretion.

Compared with the three main braided reaches A–C, the change in reach E appears to be much gentler. Reach E showed obvious changes only during the periods 1990–2000 and 1990–2020 with different degrees of erosion on the left and right banks (Figures 7b and S5a). This phenomenon is particularly prominent in the tortuous area downstream of reach E.

As for reach F, there was no significant change in the river channel morphology during 2000–2010 and 2010–2020 owing to the low activity intensity of its river course. Therefore, only the data during 1990–2000 (Figure S6) and 1990–2020 (Figure 8a) are compared and analyzed. This study found that the left and right banks of reach F also had alternative erosion, but the erosion and accretion were lower in the meandering and turning part. This phenomenon may be attributed to the alternation and damage of the main branch upstream, which led to a large amount of sediment and the alternation of accretion and erosion downstream.

During the four study periods (Figures 8b and S7), reach G showed erosion on the left bank in general. Especially in the straight reach downstream, the erosion phenomenon of reach G was more serious than that in the meandering reach upstream. This may be caused by the decrease in altitude and the increase in temperature in reach G, the decrease in permafrost, the increase in surface runoff generated by snow melting, and the backwater in the lower reaches of the Jinsha River.

Figure 9 shows the erosion, accretion, and unchanged area of the seven reaches of the Tongtian River. Except for the minor meandering reaches F and G, the unchanged areas of reaches A–E all showed a decreasing trend during 2000–2020 with a ratio of 8%, 25.2%, 30.6%, 5.3%, 5.1%, respectively. In terms of river accretion during 2000–2020, reach C decreased the most by 17.7 km$^2$ (43.4%), and reach A increased the most by 29.0 km$^2$ (219.1%). In terms of channel erosion, only the main braided reach C showed an increasing trend after 2000, with a ratio of 105.8% (increased by 13.3 km$^2$), and reaches A, B, D, E, and G all showed a decreasing trend with a ratio of 62.8%, 22.1%, 9.6%, 98.3%, and 34.5%, respectively. Among them, the eroded area of reach E decreased to 16.61 km$^2$. During the entire study period (30 years), the erosion process mainly occurred in reaches A, B, D, and E, and the accretion process mainly occurred in reaches C, F, and G.

During 1990–2020, reach F was the most stable, with an unchanged area of around 16 km$^2$ (Figure 9). The stability of reach G is the second following reach F, showing an overall state of accretion (1.0 km$^2$). The obvious changes in the Tongtian River mainly occurred in the main braided reaches, in which reaches A and B were both eroded, with an area of 4.2 km$^2$ and 5.7 km$^2$, respectively, while reach C was in an accretion state with an area of 15.6 km$^2$ (Figure 9). Results also indicate that the overall accretion–erosion area change (i.e., the difference between the accretion and erosion area) of other reaches except for reaches A–C was less than 1.7 km$^2$ (Figure 9). Therefore, it is considered that the other reaches except for reaches A–C had reached an approximately relative equilibrium between erosion and accretion.

For all the braided reaches (A–D), the total erosion, accretion, and unchanged areas were 84.1, 87. 6, and 96.5 km$^2$, respectively (Figure 9a–d). For all the meandering reaches (E–G), the total erosion, accretion, and the unchanged areas were 10.8, 10.7, and 37.4 km$^2$, respectively (Figure 9e–g). The overall accretion–erosion area change for all the braided (meandering) reaches was 3.5 (−0.1) km$^2$, indicating an overall trend of accretion (erosion). For the entire Tongtian River, the total erosion, accretion, and unchanged areas were 94.9 km$^2$, 98.3 km$^2$, and 133.9 km$^2$, respectively (Figure 9h). The overall accretion–erosion area change for the entire Tongtian River is 3.4 km$^2$, indicating an overall trend of accretion.

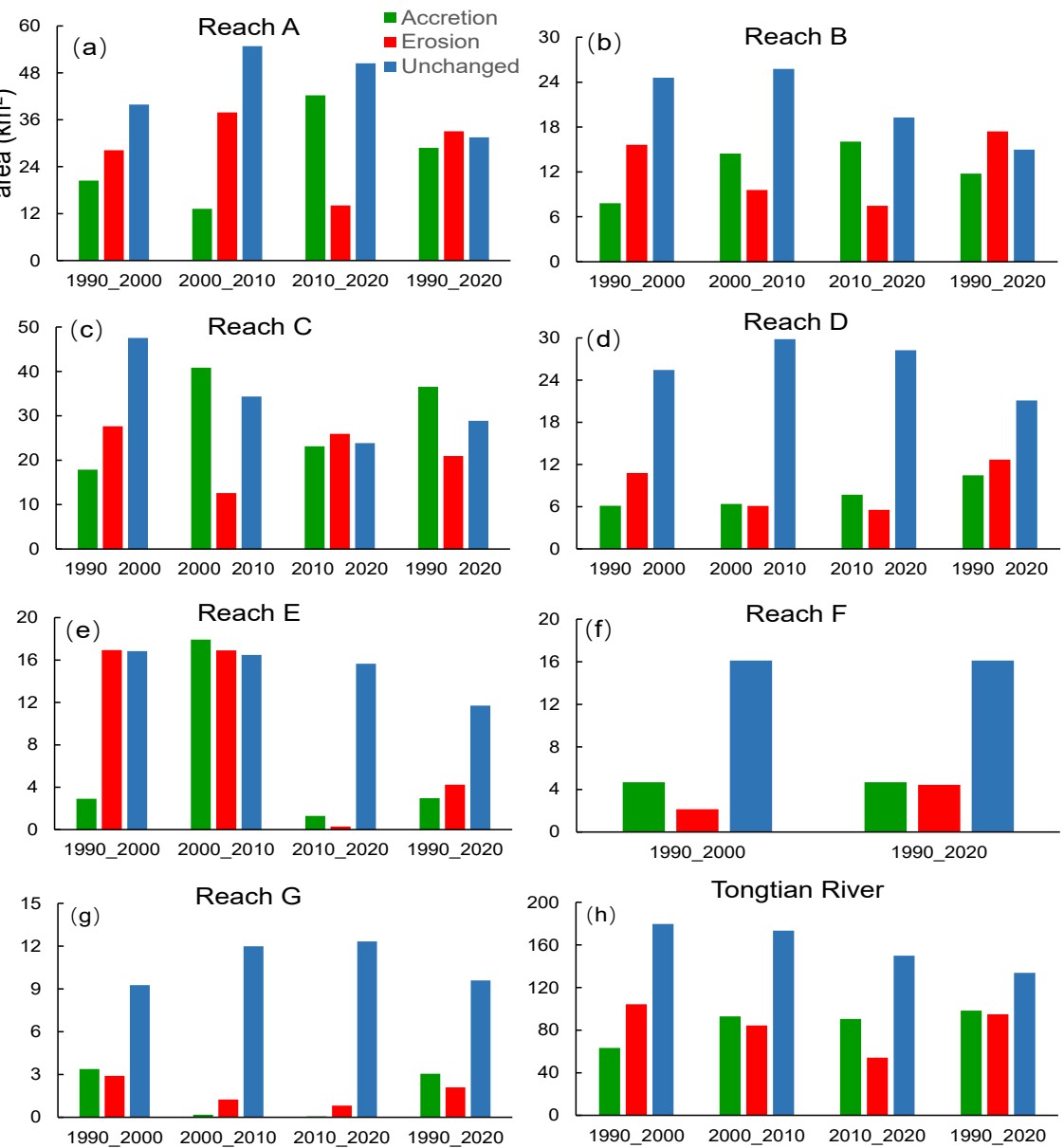

**Figure 9.** Channel lateral erosion, accretion, and unchanged area at various reaches along the Tongtian River during the four study periods since 1990. Note that the ranges of the vertical axis are different.

*3.3. Data Distribution and Correlation Analysis*

Before constructing a neural-network model, it is necessary to determine whether all variables have singular values, whether the data distribution needs to be transformed into an approximately normal distribution through a function, and whether there is an obvious simple linear relationship between each variable, to ensure the necessity and rationality of using machine learning. There are a total of eight variables for the Tongtian River channel morphology, including seven predictors (i.e., the annual averages of the sediment transport rate, streamflow, and sediment content; and the total water-body area, the accretion, erosion, and unchanged area of the river channel) and one predictand (i.e., the overall accretion-erosion area). Pearson's correlation coefficient and significance test were carried out for all of them, and the analysis results are shown in Figure 10.

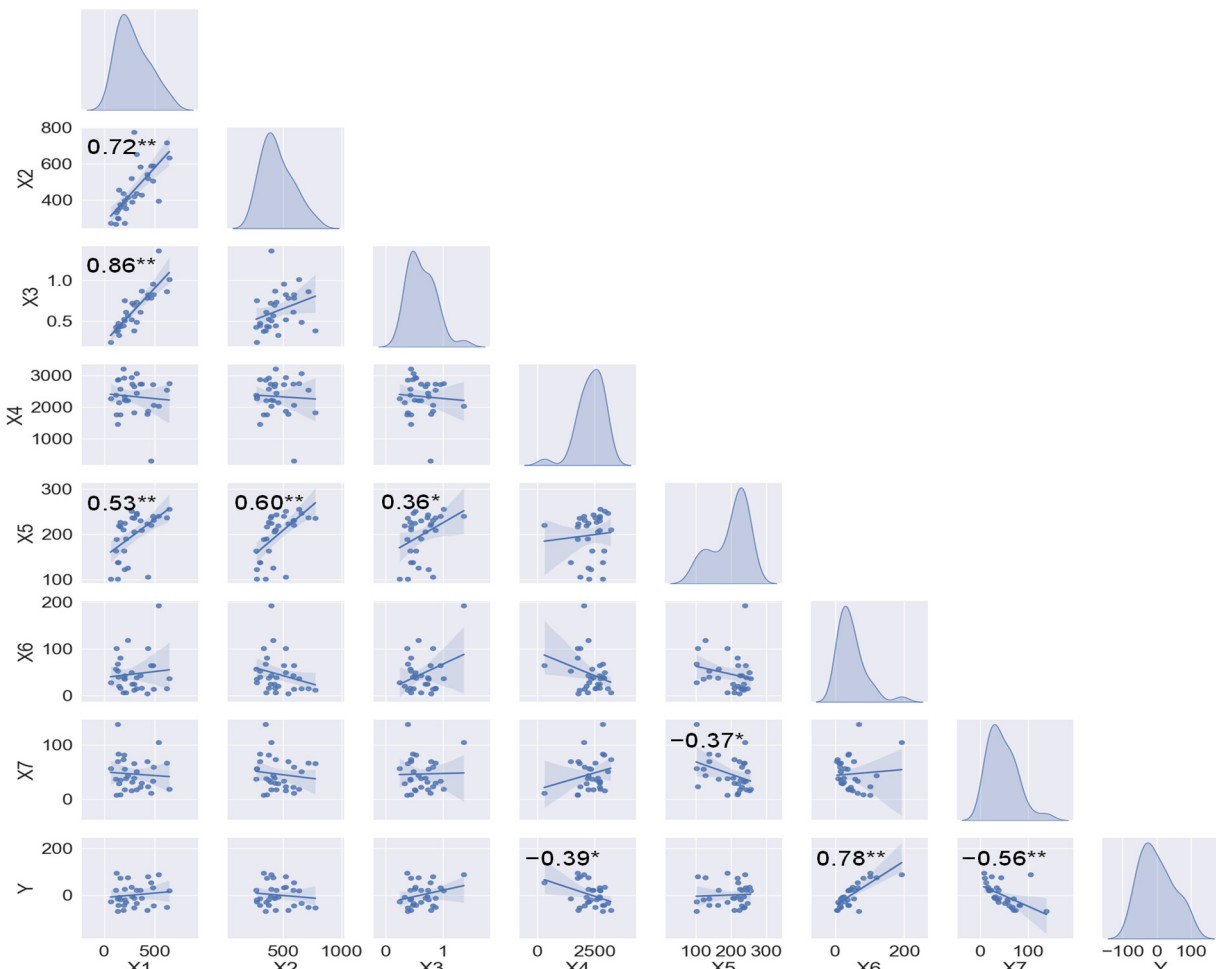

**Figure 10.** The curves are the multivariate joint distribution map and the scatterplots with a linear-fitted line show the linear correlation between any two variables of Tongtian River hydrology, sediment, and channel changes. X1 is the average annual sediment-transport rate (kg/s). X2 is the average annual streamflow (m$^3$/S). X3 is the average annual sediment content (kg/m$^3$). X4 is the water-body area (km$^2$). X5 is the unchanged area of the river (km$^2$). X6 is the accretion area (km$^2$). X7 is the erosion area (km$^2$). Y is the overall accretion–erosion area (km$^2$). The symbols "**" and "*" indicate that the correlation coefficient has passed the significance test at the level of 0.99 and 0.95 respectively. Those scatterplots without marking correlation coefficients indicate an insignificant correlation.

The variables in this study are approximately normally distributed. Among the eight variables, the data distribution curves for the annual average sediment-transport rate and the annual average flow are the smoothest and the closest to the standard normal distribution. Among all variables, the overall accretion–erosion area of the Tongtian River is significantly linearly correlated with the accretion area ($r = 0.78$) and erosion area ($r = −0.56$) and is significantly linearly correlated with the water-body area ($r = −0.39$) (Figure 10). The annual average sediment-transport rate has a very significant linear correlation with the annual average streamflow ($r = 0.72$), the annual average sediment concentration ($r = 0.86$), and the unchanged area of the river channel ($r = 0.53$) (Figure 10). There is a very significant linear correlation between the unchanged area of the river channel and the annual average streamflow ($r = 0.60$). Furthermore, the unchanged area of the river channel has a significant linear correlation with the annual average sediment content ($r = 0.36$) and the erosion area ($r = −0.37$) (Figure 10).

To sum up, there is a certain linear relationship between some variables, but there is an implicit nonlinear relationship between more variables. If only a simple linear correlation is used to reveal the relationship between each hydrological variable and the overall accretion–erosion area change of the Tongtian River, a large number of mapping relationships, implicit relationships and interactive relationships will be missed. Therefore, machine learning is used to further mine, extract, and analyze the relationship between the hydrological variable and the overall accretion–erosion area change, and describe the change law of river-channel morphology.

### 3.4. Predictive Performance of the Lightweight Neural-Network Model

The lightweight artificial-intelligence model (epoch: 200, learning rate: 0.003, the number of neurons in the first (second) hidden layer: 10 (5)) was constructed to dig deeply into the relationship between hydrological variables and the overall accretion–erosion area change. This method can more intelligently and deeply understand and predict the plane geometric characteristic of the Tongtian River. It can be seen from Figure 11a that the loss function of the model (training session) drops rapidly and starts to converge at about 85 epochs, and eventually tends to be smooth without obvious oscillations, and the value of MSE is very close to 0 at 200 epochs, indicating that the error between predicted and the actual values is very small. At 200 epochs, the optimal lightweight intelligent model for predicting the overall accretion–erosion area of the Tongtian River is obtained. Figure 11b shows the annual total accretion–erosion change (test session) in the Tongtian River Basin during the study period. The negative number indicates erosion, and the positive number indicates accretion. Overall, the predicted value of the accretion–erosion area change is very close to the actual value with a correlation coefficient close to 1.0 and the MSE of 4.6 km$^2$ (Figure 11b). Among the nine randomly selected test groups, only the second, fifth, and eighth groups are significantly different from the true value, with an overestimation of the overall accretion–erosion area by 3.3, 3.7, and 3.9 km$^2$, respectively (Figure 11b). These obvious differences may be due to insufficient hydrological feature vectors considered in this study and no consideration of meteorological features, channel bed slope, and particle-size distribution in the bed materials.

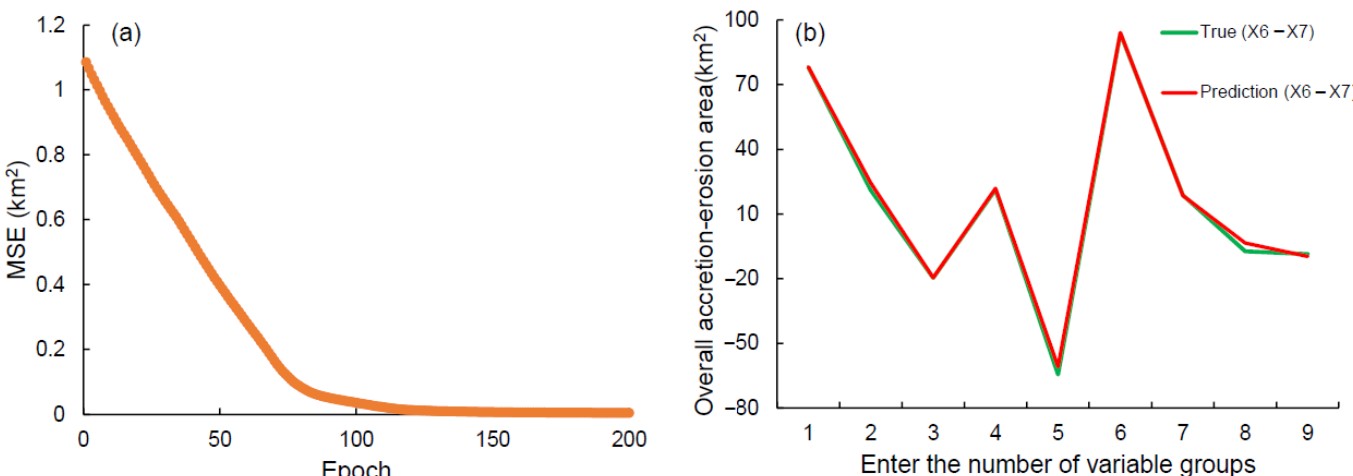

**Figure 11.** (**a**) Statistical performance of a lightweight smart model showing MSE between predicted and true values based on the training sets. (**b**) Comparison of the predicted overall accretion–erosion degree (variable X6 minus X7) of the Tongtian River with the true value using a lightweight intelligent model based on the test sets.

As shown in Figure 12, although the predicted values are obtained, and a linear fit is performed between each input feature variable and the predicted value, it is expected to obtain an obvious linear relationship between them. However, there is no obvious linear relationship (insignificant correlation lower than 0.4) and regularity between each hydro-

logical variable and the overall accretion–erosion area of the Tongtian River, indicating that the overall accretion–erosion area change of the Tongtian River is determined by the synergistic effect of various variables. Through the lightweight intelligent prediction model, the implicit relationship between each variable and the overall accretion–erosion degree can be deeply excavated, and the real situation can be better reproduced.

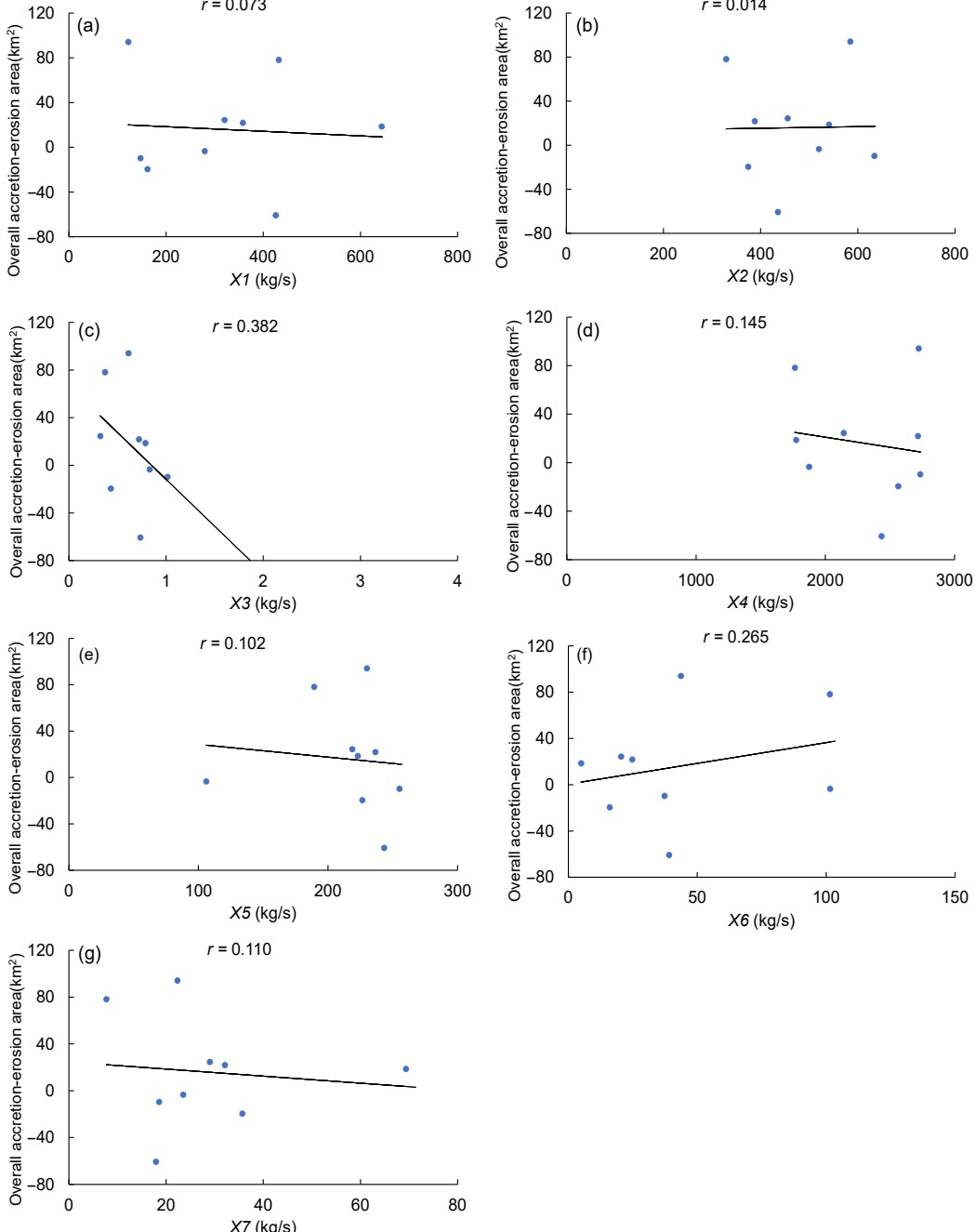

**Figure 12.** Relationship between each hydrological variable and the overall accretion–erosion area of the Tongtian River. X1 is the average annual sediment-transport rate (kg/s). X2 is the average annual streamflow ($m^3$/S). X3 is the average annual sediment content (kg/$m^3$). X4 is the water-body area ($km^2$). X5 is the unchanged area of the river ($km^2$). X6 is the accretion area ($km^2$). X7 is the erosion area ($km^2$). The black line is the linearly fitted line between the single-input hydrological variable and the prediction value of the overall accretion–erosion area of the Tongtian River. *r* is the correlation coefficient of the two variables. All the correlation coefficients do not pass the significant test.

## 4. Discussion

### 4.1. Factors Influencing River-Channel Morphology

Factors that cause changes in the river course are multisource and multi-interaction. In addition to external factors, changes in the river course are related to the sediment characteristics and hydraulic characteristics of the river itself [47–49]. The morphology of a river can be adjusted by the flow and sediment transport rate [50]. The erosion and accretion of the river channel are caused by the synergy of water and sediment.

The significant correlations between the annual averages of the sediment-transport rate, streamflow, and sediment concentration and the unchanged area of the river channel (see Figure 10 and Section 3.3) may be owing to the fact that the water and sediment conditions control the main flow direction of the river and determine the location of the river center line. The significant correlation between the unchanged area of the river channel and the erosion area (see Figure 10 and Section 3.3) may be due to the fact that the annual average sediment concentration controls the balance between erosion and accretion of the river channel, and the erosion of the river channel will increase the annual average sediment concentration. The increased annual average sediment concentration will further raise the riverbed and change the river channel to form accretion under the condition of constant flow, thus forming a water-sediment synergy. Consequently, the overall accretion–erosion area of the Tongtian River is significantly linearly correlated with the accretion area ($r = 0.78$) and erosion area ($r = -0.56$). Therefore, it can be seen that the Tongtian River is generally accreted, which may be due to the enhanced sediment-carrying capacity of surface water caused by the surface runoff generated by snow melting, and more sediment flowing into the Tongtian River through other tributaries.

Some other factors also influence the morphology of the Tongtian River. Global warming has reduced the proportion of permafrost and degraded the glaciers in the source area of the TRSR, making it easier for the soil in the TRSR to flow into the Tongtian River under the influence of rainfall and snow melting. In addition, the East Asian monsoon increases the temperature in the source region of the Yangtze River [51], resulting in the melting of glaciers and the formation of more surface runoff, which further carries surface sediment into the river, causing the Tongtian River to accrete. Moreover, the increased dry deposition of atmospheric particles driven by the winter monsoon leads the rivers to acquire more sediment particles and form accretion [52,53].

### 4.2. Characteristics and Formation of Braided River Courses of the Tongtian River

The upper braided reaches of the Tongtian River are more active than the lower meandering reaches. In this study, the braided river course in the upper reaches of the Tongtian River has frequent and continuous alternation, fragmentation, and reorganization of main branches and secondary branches, which carry a large amount of water and sands and migrate downstream to make reach C the main accretion area of the Tongtian River. The right source of the Yangtze River, Dangqu, and its west source, Ulan Moron, converge to form the Tongtian River. The braided reaches A and B in the upper reaches of the Tongtian River are both close to the confluence of Dangqu and Ulan Moron. It is still unclear whether the confluence affects the alternation of main branches and the secondary branches of the braided river course because there is limited information on the effect of the upstream river confluence on the accretion of the downstream reach C. Future studies are required to take a deeper insight to the above issue as well as the question on whether the plane geometries of the three braided river courses are related to each other.

### 4.3. Problems Associated with the JRC Dataset

For surface-water distribution projects, more researchers recently preferred to use the JRC dataset for water-resource-management research [54–56]. The JRC dataset has excellent performance in research on the transparency change of water quality in Lake Taihu [57], the transparency of global lake water [58], and dynamic changes of lakes in the Mongolian Plateau [32]. This study adopted the JRC's annual water-body dataset and used the expert

system to classify each pixel individually as water/non-water. Permanent water bodies, seasonal water bodies, and non-water bodies are also covered in this dataset. Due to the Tongtian River being located in the TRSR, there is a lot of snow in winter. When the snow melts, it will generate a lot of surface runoff, and the phenomenon of snow melting will exist for a long time. Therefore, the water bodies we extracted are all water-body elements occurring three or more times per pixel in a year, thereby reducing the impact of surface runoff caused by snowmelt on the early results of water bodies. However, we found that the JRC dataset cannot reproduce the complex geometric shape of the braided channels of the Tongtian River as indicated in Figure 3. This may be because that the secondary branches and the main branch have strong activities in a short time, and there are a lot of alternations and damages. This behavior can persist for a long time, so when extracting water bodies using JRC's annual dataset, the water-body information of the main branch and secondary branches will be missed. However, in the lower reaches of the Tongtian River (reaches E–G), the JRC dataset can reproduce the channel morphology more comprehensively and completely, and the water-body area is closer to the real value, and the remote-sensing water image is smoother.

*4.4. Evaluation of the Lightweight Neural-Network Prediction Model*

In the prediction modeling of the Tongtian River morphology, at 200 Epochs during the training session, the MSE is 0.0048 km$^2$, indicating that the average error between the model predicted and the true area of overall accretion–erosion is small. The error is negligible compared to the basin area of the Tongtian River (137,700 km$^2$). The neural-network model constructed in this study is very simple in structure, with only two hidden layers. If the number of neurons is not considered, logistic regression in statistics can be regarded as the simplest neural network. Our model only has one more hidden layer than logistic regression and adds neurons to the hidden layer. The "Relu" activation function was used to solve the nonlinear mapping relationship, and the "Adam" optimization function was used to solve the weight and bias of the membership function. While keeping the model structure simple, the model performance was also improved to better reproduce the complex nonlinear processes in the changes in river morphology.

Artificial neural-network models can implicitly identify complex nonlinear connections between independent and dependent parameters and can detect all potential interactions across predictor parameters [38,59]. In the prediction of the overall accretion–erosion area of the Tongtian River, the choice of the "best" prediction model is a compromise between the accuracy of model prediction and the complexity of the model. The main advantages of the lightweight neural-network model developed in this study are the simplicity of the model and the low running cost, with little impact on the model performance. In many relevant studies, the prediction of river morphology is usually limited by the availability of river hydraulic and hydrological data, so it is difficult to develop complex artificial-intelligence models. However, the superior performance and simple structure of the model in this study have great potential for application in important data-scarce basins, especially in developing countries, where there may be a lack of technical modeling skills and understanding of the hydraulic and sedimentary dynamics in river systems.

**5. Conclusions**

This study investigated the spatiotemporal variability of river-channel morphology of the Tongtian River with a focus on monitoring and predicting the erosion, accretion, and unchanged areas during the past 30 years (1990–2020). The water bodies of the Tongtian River were extracted using Landsat images based on the combined method of three water-body indices and a threshold, and the JRC global surface-water dataset. Results indicate that the combined method is the most accurate for water-body extraction for the braided reaches compared to the JRC dataset and the single water-body index such as the *NDVI* and *EVI*, while the JRC dataset performs better than other methods for the meandering reaches.

For the entire Tongtian River, the total accretion, erosion, and unchanged area are 98.3, 94.9, and 133.9 km$^2$, respectively, during the past 30 years. The difference between the total accretion and erosion area is 3.4 km$^2$, indicating an overall trend of accretion for the Tongtian River channel morphology. Among the seven reaches of the Tongtian River, braided reach C experienced the most accretion (36.5 km$^2$), while braided reach A experienced the most erosion (33.1 km$^2$) during the past 30 years. The channel erosion and accretion in the Tongtian River are determined by the synergistic effects of sediment transport rate and streamflow.

The lightweight neural-network model was built to predict the overall accretion–erosion area of the Tongtian River. The most intelligent model was obtained after training for 200 epochs. The MSE between the predicted value and the actual value is 0.0048 km$^2$ for the training session, and 4.6 km$^2$ for the test session. Result proves that the model can well reproduce the complex nonlinear processes in the morphology change of the river by simplifying the model structure.

The approach for extracting water bodies and analyzing and predicting the channel morphology can be applied to other rivers, especially those with braided features. Future research on the complex internal causes of river morphological changes may further consider other factors such as hydrodynamic factors, geometric characteristics of the braided river course, and the influence of other rivers' confluences.

**Supplementary Materials:** The following supporting information can be downloaded at: https://www.mdpi.com/article/10.3390/rs14133107/s1. Table S1. Paths and rows of satellite images covered by Landsat in the study region. Table S2. Product description of remote-sensing data. Figure S1. Maps showing channel morphology dynamics for the main braided reach A of the Tongtian River during three periods, i.e., 1990–2000 (a), 2000–2010 (b), 2010–2020 (c). Figure S2. The same as Figure S1, but for the main braided reach B. Figure S3. The same as Figure S1, but for the main braided reach C. Figure S4. The same as Figure S1, but for the minor braided reach D. A local part highlighted in a colored rectangle is enlarged in an inserted map with a rectangle in the same color. Figure S5. The same as Figure S4, but for the main meandering reach E. Figure S6. The same as Figure S4, but for the main meandering reach F. Only the map during 1990–2000 is shown here. Figure S7. The same as Figure S4, but for the minor meandering reach G.

**Author Contributions:** Methodology, revision of the manuscript, Z.H.; data curation, W.L. and Y.X.; writing—original draft preparation, K.X.; writing—review and editing, B.D.; visualization, J.L.; supervision, C.J. All authors have read and agreed to the published version of the manuscript.

**Funding:** This research was funded by the National Natural Science Foundation of China [grant number 51979015 and 91647118], and partial support comes from the National Science Foundation of Hunan Province, China (grant number 2021JJ30707), the Science and Technology Innovation Program of Hunan Province, China (grant numbers 2020RC3037 and 20hnkj019).

**Data Availability Statement:** This study uses the GEE platform, which can be accessed here: https://earthengine.google.com (accessed on 1 December 2021). The platform also includes the used Landsat series of remote-sensing images. The hydrological data obtained from the "China Sediment Bulletin" in this study can be found here: http://www.mwr.gov.cn/sj/tjgb/zghlnsgb/ (accessed on 20 December 2021).

**Conflicts of Interest:** The authors declare no conflict of interest.

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
