# Peer review of "Monitoring and Predicting Channel Morphology of the Tongtian River, Headwater of the Yangtze River Using Landsat Images and Lightweight Neural Network"

_remotesensing, doi:10.3390/rs14133107_

Round 1

Reviewer 1 Report

General Comments:

This manuscript focused on river course changes using satellite images and JRC data base. In addition, an intelligent light-weight neural network model was constructed to predict and analyze the overall scouring in river reaches of the Tongtian River. As the analysis results presented in the manuscript, the research topic is interesting that may enhance our understanding in river morphological changes, and I think the research topic is also within the scope of the journal. However, the authors should further improve their manuscript, and also manuscript needs some clarification before consideration for publication. Furthermore, authors should clearly describe new findings in their study than previous studies as authors have also pointed out that few studies have already studied the channel morphology of the Tongtian River using remote sensing, GIS techniques and the artificial intelligence methods. The specific comments in detail are listed below.

Specific Comments:

1.      Lines 29-30, readers may not understand the characteristics of each reach (A-G) just by reading abstract. So, for better understanding, I suggest to describe the descriptions on occurrence of erosion and sedimentation based on characteristics of reach or channel types, etc.

2.      Introduction section, overall, I suggest to further improve introduction section by re-structuring the descriptions and deleting the repeated descriptions.

3.      Line 53, what are the three main channel types? Please describe these.

4.      Lines 60-62, this study also limited to river reaches. In addition, please also clarify why global analysis and multi-temporal research on braided rivers from a broader and larger perspective are crucial.

5.      Lines 100-114, these descriptions could be moved to the study area section.

6.      Lines 115-116, authors mentioned that few studies have studied the channel morphology of the Tongtian River using remote sensing, GIS techniques and artificial intelligence methods. However, it is not clearly understandable that what are the gaps in those previous study and what are the improvements in this study. In addition, what are the improvements or new findings of this study than those previous studies should be clearly described in results or discussions section.

7.      Lines 124-125, I suggest to clary the reasons for dividing the study period 1990-2020 into 3 decadal periods (1990-2000, 2000-2010, 2010-2020). Are there any specific reasons for dividing the study period into 4 episodes?  For example, change point in the data or other reasons?

8.      Line 139, Yangtze River à Tangtian River??

9.      Figure 1 appeared in the middle of the Figure Caption.

10.  In Figure 1, please show the location map of Tongtian River in Asia or China. I also suggest to show the river network of Jinsha and Yangtze Rivers.

11.  Lines 170-171, what are the criteria to divide main braided reach or secondary braided reach or main meander reach or minor meander reach? Please describe criteria/definition or add references.

12.  Lines 190-192, please clarify that what approach/method was used to repairs / improve the accuracy of Landsat data or add references.

13.  Lines 205-209, these descriptions are completely repeated.

14.  Line 209, from 1990 to 2021 à from 1990 to 2020 ?? (Figure 2 shows from 1990 to 2020)

15.  Line 212, I suggest to add reference or clarify on using threshold value 0.1 for detecting water body.

16.  Line 253, please clarify that why the distance should be 50m.

17.  Line 261. This sentence should be clearly described. For example, criteria for recording/detecting the Tongtian River should be clarified.

18.  Lines 269-271 and Figures 3c and 3d; the water channels in the figures 3c and 3d are not clearly visible, even in the enlarge view of the white part. So, it is hard to say how JRC is better, how the channel is smoother. Therefore, I suggest to improve the quality of Figs. 3c and 3d. Authors may show the enlarge views of several small parts.

19.  In table 3, main braided reach à main braided reach 1 ??

20.  Lines, 336-337, please clarify that why only hydrological data from 2014 to 2020 were used to train the model. Is the 7 years data enough to train the model?

21.  Line 354, please define the full form of Adm and Relu.

22.  Line 368, Figures above show??

23.  The channel morphology for reaches A, B, C, D, and E presented in Figs 5-9, respectively, do not correspond to channel morphology/reach network of each reach presented in Figure 1. The shape of channel and lat/long coordinates of the river reaches are completely different than those in Figure 1. For example, channel morphology for reach A presented in Fig. 5 seems the channel morphology for upstream part of reach A (upstream part of point A), not for reach A. Please check the figures, and correct these including the descriptions.

In addition, the quality of Figs. 5-11 should be improved, particularly Figs. 8-11. It is hard to identify unchanged or accretion or erosion areas in these figures. Authors may use enlarge view of the figure for important part. Furthermore, quality of alpha-numerical characters should be also improved in all the figures.

24.  Lines 433-435, what could be the possible reasons?

25.  Lines 523-524, is only 9 randomly selected sets enough for testing and accuracy assessment? How closure is the predicted value to the actual value? It would be good if authors could use some quantitative metrices for accuracy assessment.

26.  Lines 525-527, how’s about the difference between the predicted and actual values also due to no consideration of channel bed slope, particle size distribution in the bed materials?

27.  Lines 551-569, these descriptions are too general and qualitative. I suggest to improve these incorporating findings from the results and analysis.

28.  In conclusion, I suggest to describe the conclusion section much more concisely. Currently authors just listed the descriptions from the results.

29.  Line 642, “, the source of Yangtze River,” could be deleted.

Author Response

Reviewer #1

Comments and Suggestions for Authors

General Comments:

This manuscript focused on river course changes using satellite images and JRC data base. In addition, an intelligent light-weight neural network model was constructed to predict and analyze the overall scouring in river reaches of the Tongtian River. As the analysis results presented in the manuscript, the research topic is interesting that may enhance our understanding in river morphological changes, and I think the research topic is also within the scope of the journal. However, the authors should further improve their manuscript, and also manuscript needs some clarification before consideration for publication. Furthermore, authors should clearly describe new findings in their study than previous studies as authors have also pointed out that few studies have already studied the channel morphology of the Tongtian River using remote sensing, GIS techniques and the artificial intelligence methods. The specific comments in detail are listed below.

Response: Thank you very much for your professional evaluation and insightful comments. We have revised the manuscript carefully following the reviewers’ comments. The major revisions include the rewriting of the Abstract, Introduction, Conclusion and some parts of Results and Discussion. In the Introduction, we have made a more comprehensive literature review on the channel morphology in the Tongtian River. The research objectives of the revised manuscript are much clearer, and some new findings and conclusions were summarized. We further added a sub-section in the Results and Discussion to assess the accuracy of different water body extraction methods. We believe that the revised manuscript is of much higher quality than the previous version. Please find the point-by-point response to the reviewers’ comments below. The major revisions were highlighted in yellow in the manuscript file named “changes highlighted”.

Specific Comments:

  1. Lines 29-30, readers may not understand the characteristics of each reach (A-G) just by reading abstract. So, for better understanding, I suggest to describe the descriptions on occurrence of erosion and sedimentation based on characteristics of reach or channel types, etc.

Response: Agree. The Abstract has been improved and does not include the ambiguous information on reach A-G. In the improved Abstract, we presented the major findings on the total accretion and erosion area of the Tongtian River and the trend of accretion or erosion of different breaches, instead of focusing on an individual reach. Please see Lines 17-30, Page 1 for details.

  1. Introduction section, overall, I suggest to further improve introduction section by re-structuring the descriptions and deleting the repeated descriptions.

      Response: Agree. We have rewritten the Introduction. Firstly, we reviewed the studies on channel morphology of the Tongtian River and found that previous studies seldom focused on (1) the erosion and accretion processes of the Tongtian River channel morphology at the large spatial scale and multi-temporal scales and (2) the prediction of the channel morphology of the Tongtian River. This study aims at bridging these two research gaps. Secondly, we reviewed the studies on different water body extraction methods and found that it is still unclear in previous studies whether the accuracy of the JRC global surface water dataset is better than the single water body index and the combined method of multiple water body indices. This is also a promising topic to be studied. Thirdly, a literature review was made for the investigation of the evolution of river channel morphology based on machine learning. We found that few studies have used the lightweight neural network (a simpler but more efficient method) to predict the accretion and erosion of the river channel morphology, which is another research objective of this study. This study intends to answer the following two key scientific questions: (1) What are the spatio-temporal characteristics, trends and associated influencing factors of the accretion and erosion area change of the Tongtian River channel morphology? (2) Can the lightweight neural network model well predict the accretion and erosion area change of the Tongtian River channel morphology, and how is the prediction accuracy? Overall, the revised Introduction is much clearer and more logical on the literature review, research objectives, scientific questions, as well as the methodology. Please see Lines 37-158, Pages 1-4 for more details.

  1. Line 53, what are the three main channel types? Please describe these.

      Response: We thank the reviewer for this comment. Due to the rewriting of the entire Introduction, previous descriptions on the three main channel types have been deleted.

  1. Lines 60-62, this study also limited to river reaches. In addition, please also clarify why global analysis and multi-temporal research on braided rivers from a broader and larger perspective are crucial.

      Response: Although we also focused on reach scale, all the reaches for the entire Tongtian River were studied. We did not focus on a single reach. In the revised manuscript, we further quantified the accretion and erosion area of the entire Tongtian River, which was neglected in the previous manuscript. Also, we further made a literature review on channel morphology of the Tongtian River and found that previous studies seldom focus on the erosion and accretion processes of the Tongtian River channel morphology at the large spatial (or global) scale and multi-temporal scales. This study aims at bridging this research gap. A global-scale study can reveal the spatial differences on channel morphology, which is of great significance for river course management. A multi-temporal study can reveal the dynamic changes of channel morphology, which are important for a better understanding of the historical and recent status, and future trends of channel morphology. Therefore, a global analysis and multi-temporal research is crucial.

  1. Lines 100-114, these descriptions could be moved to the study area section.

      Response: The Introduction has been totally rewritten. Most of the descriptions of the study area in the previous manuscript have been deleted. We only briefly described the study area (the Three-River-Source Region and the Tongtian River) in the first paragraph of the Introduction. Please see Lines 37-49, Pages 1-2 for details.

  1. Lines 115-116, authors mentioned that few studies have studied the channel morphology of the Tongtian River using remote sensing, GIS techniques and artificial intelligence methods. However, it is not clearly understandable that what are the gaps in those previous study and what are the improvements in this study. In addition, what are the improvements or new findings of this study than those previous studies should be clearly described in results or discussions section.

      Response: In the revised manuscript, we further reviewed the studies on channel morphology of the Tongtian River and find that previous studies seldom focus on the erosion and accretion processes of the Tongtian River channel morphology at the large spatial scale and multi-temporal scales. This study bridged this research gap. Also, the methods to extract water body and predict river channel morphology are different from previous studies. The major improvements are (1) we selected the optimal method for water body extraction from several different methods, (2) the simpler and more efficient lightweight neural network model is used for predicting the river channel morphology. We added one more sub-section (Section 3.1) in the Results and Discussion to assess the accuracy of different water body extraction methods. Some new findings on the total erosion and accretion area change of the entire Tongtian River were summarized in the Results and Discussion.

  1. Lines 124-125, I suggest to clary the reasons for dividing the study period 1990-2020 into 3 decadal periods (1990-2000, 2000-2010, 2010-2020). Are there any specific reasons for dividing the study period into 4 episodes?  For example, change point in the data or other reasons?

      Response: According to the fact that the Tongtian River has a relatively small change range compared with other rivers in the plain areas, the dynamic changes of the river channel morphology cannot be identified from the differences between the satellite images in two close periods. We referred to the references (e.g., Chen et al. 2021 and Hiraga et al.2018) to divide the study period 1990-2020 into the three decadal periods. Results show that the channel morphological changes of the Tongtian River can be well detected at the decadal time scale. Therefore, the division of the three periods is rational.

  1. Line 139, Yangtze River à Tangtian River??

      Response: “Yangtze River” has been revised to “Tongtian River”. Please see Line 164, Page 4.

  1. Figure 1 appeared in the middle of the Figure Caption.

Response: We have rearranged the figure and put the title of the figure at the bottom of the figure.

  1. In Figure 1, please show the location map of Tongtian River in Asia or China. I also suggest to show the river network of Jinsha and Yangtze Rivers.

      Response: Revised as suggested. We have updated the Figure 1 by inserting a location map for the Tongtian River. Also, the major rivers of the Yangtze River (including the Jinsha River) are also shown in the location map. The updated Figure 1 is shown in Lines 209-214, Page 5.

  1. Lines 170-171, what are the criteria to divide main braided reach or secondary braided reach or main meander reach or minor meander reach? Please describe criteria/definition or add references.

      Response: According to the plane morphological characteristics, we divided the river channel into braided reaches (A, B, C, and D) and meandering reaches (E, F, and G). Among the braided reaches, the reaches A, B, and C show high intensity of morphological changes and are classified as main braided reaches, while reach D shows weak intensity of morphological changes, and is classified as minor braided reach. Among the meandering reaches, the reaches E and F show obvious morphological characteristics and are classified as main meandering reach, while the reach G shows relatively insignificant morphological characteristics and is classified as the minor meandering reach. This part of description was added in the revised manuscript. Please see Lines 314-321, Page 9.

  1. Lines 190-192, please clarify that what approach/method was used to repairs / improve the accuracy of Landsat data or add references.

Response: In order to reduce the impact of fringe noise in Landsat-7 images and improve the accu-racy of water body extraction from all remote sensing images, the water body images extracted from May to October each year through the combined method are synthesized using the reducer module in GEE to form an inter-annual water body image. The operation procedure is as follows: Sum up the pixel values in the same grid in all water images. When all pixels in the grid are summed, a composite image can be obtained. This part of description was added in the revised manuscript. Please see Lines 246-251, Page 7 for details.

  1. Lines 205-209, these descriptions are completely repeated.

      Response: Agree. This part of information has been deleted.

  1. Line 209, from 1990 to 2021 à from 1990 to 2020 ?? (Figure 2 shows from 1990 to 2020)

      Response: According to the last comment, this part of repeated information has been deleted.

  1. Line 212, I suggest to add reference or clarify on using threshold value 0.1 for detecting water body.

      Response: Agree. References has been added for the threshold value 0.1. Please see Line 239, Page 6.

  1. Line 253, please clarify that why the distance should be 50m.

      Response: To extract the water bodies more accurately, a buffer zone of 50 m is set up along the edge of river channel to cover all the water bodies of the river channel. The buffer zone is set to be larger than the resolution (30 m) of the Landsat images and less than the distance of 2 pixels (60 m). This buffer zone is reasonable because a larger buffer zone will increase the computation burden for water body identification and classification, and a smaller buffer zone with only one pixel (30 m) is unable to encompass the water body with a distance further than 30 m to the edge of river channel. This part of description was added in the revised manuscript. Please see Line 252-265, Page 7.

  1. Line 261. This sentence should be clearly described. For example, criteria for recording/detecting the Tongtian River should be clarified.

      Response: This sentence has been revised to be more clearly as follows: According to the new feature attribute, if the number of times of the water body information in a single pixel of the 6 scene images are ≥ 3 times, that single pixel is recorded as the Tongtian River. Please see Lines 277-280, Page 7.

  1. Lines 269-271 and Figures 3c and 3d; the water channels in the figures 3c and 3d are not clearly visible, even in the enlarge view of the white part. So, it is hard to say how JRC is better, how the channel is smoother. Therefore, I suggest to improve the quality of Figs. 3c and 3d. Authors may show the enlarge views of several small parts.

      Response: Agree. We redrew Figure 3. The new figure with enlarged views is much clearer to show the differences between the water bodies extracted from the combined method and the JRC dataset. Please see Line 306, Page 8.

  1. In table 3, main braided reach à main braided reach 1 ??

      Response: We originally wanted to use the numbers 1-3 to further distinguish the three main braided reaches, but these numbers are to some extent redundant. Hence, we deleted these numbers to make the channel type more clearly. Please see Line 335, Page 9.

  1. Lines, 336-337, please clarify that why only hydrological data from 2014 to 2020 were used to train the model. Is the 7 years data enough to train the model?

      Response: The machine learning model used (1) 30-year annual average hydrological data of sedi-ment transport rate, streamflow, and sediment content obtained from Zhimenda Hydrological Station (1990-2013) (97°14'18''E, 33°00'46''N, Figure 1) managed by the Qinghai Water Conservancy Department and the Bulletin of China River Sediment provided by the Ministry of Water Resources of China (2014-2020), and (2) total water body area, the accretion area, erosion area, and unchanged area obtained through the above-mentioned remote sensing image processing procedures. The relevant sentences have been revised to be more clearly. Please see Lines 366-373, Page 10 for details.

  1. Line 354, please define the full form of Adm and Relu.

      Response: The “Adam” and “Relu” are embedded functions in the tensorflow module in Python, which can be directly called. We mentioned this information in the revised manuscript. Please see Line 387, Page 10.

  1. Line 368, Figures above show??

      Response: We have re-designed all the Figures 5-11 in the previous manuscript. Now, Figures 6-8 show the dynamics of historical river erosion and accretion at seven selected reaches during 1990-2020. The results for the three decadal episodes (1990-2000, 2000-2010, 2010-2020) are shown in Figures S1-S7. The citations for the figures were updated.

  1. The channel morphology for reaches A, B, C, D, and E presented in Figs 5-9, respectively, do not correspond to channel morphology/reach network of each reach presented in Figure 1. The shape of channel and lat/long coordinates of the river reaches are completely different than those in Figure 1. For example, channel morphology for reach A presented in Fig. 5 seems the channel morphology for upstream part of reach A (upstream part of point A), not for reach A. Please check the figures, and correct these including the descriptions.

In addition, the quality of Figs. 5-11 should be improved, particularly Figs. 8-11. It is hard to identify unchanged or accretion or erosion areas in these figures. Authors may use enlarge view of the figure for important part. Furthermore, quality of alpha-numerical characters should be also improved in all the figures.

Response: We thank the reviewer for this detailed comment. (1) We have corrected the mistakes in the overview map of the study area. A location map of the Tongtian River was also inserted in Figure 1 according to other reviewers’ comments. (2) We have re-designed all the Figures 5-11 in the previous manuscript. Now, Figures 6-8 show the dynamics of historical river erosion and accretion at seven selected reaches during 1990-2020. The results for the three decadal episodes (1990-2000, 2000-2010, 2010-2020) are shown in Figures S1-S7. The updated figures are of higher resolution and have clearer enlarged views. Please see the updated Figures 6-8 in the manuscript and the supplementary materials for details.

  1. Lines 433-435, what could be the possible reasons?

      Response: This phenomenon may be attributed to that the alternation and damage of the main branch of the upstream lead to the large amount of sediment and the alternation of accretion and erosion in the downstream. Please see Line 494-498, Page 16 for this part of analysis.

  1. Lines 523-524, is only 9 randomly selected sets enough for testing and accuracy assessment? How closure is the predicted value to the actual value? It would be good if authors could use some quantitative metrices for accuracy assessment.

      Response: The intelligent lightweight prediction model randomly selects the training set and the test set according to the ratio of 7:3 following previous studies (i.e., Beyaztas et al.2021, Jamil et al.2022 and Huang et al.2021). As there are 30 years (or sets) of data, the number of test sets is 30%*30=9. Therefore, the 9 sets are enough for testing and accuracy assessment. This study used two quantitative metrices for accuracy assessment, i.e., Pearson’s correlation coefficient and the mean square error (MSE) (Please see equations (4) and (5) in the revised manuscript). Based on these two metrices, the predicted value of accretion-erosion change area is very close to the actual value with a correlation coefficient close to 1.0 and the MSE of 4.6 km2. Among the 9 randomly selected test groups, only the second, fifth and eighth group are significantly different from the true value, with an overestimation of 3.3, 3.7, and 3.9 km2 (Figure 11b). This part of result analyses was added in the revised manuscript. Please see Line 594-601, Page 19.

  1. Lines 525-527, how’s about the difference between the predicted and actual values also due to no consideration of channel bed slope, particle size distribution in the bed materials?

      Response: Agree. We have mentioned these factors in the revised manuscript. Please see Line 600, Page 19.

  1. Lines 551-569, these descriptions are too general and qualitative. I suggest to improve these incorporating findings from the results and analysis.

      Response: We rewrote the Lines 551-569 in the previous manuscript by incorporating findings from the results and analysis. We further discussed the reasons for the relationships between the annual averages of the sediment transport rate, discharge, and sediment concentration and the unchanged area, and between the unchanged area and erosion area of the river channel. Also, we also further discussed the influences of other factors such as global warming, East Asian monsoon, and the increased dry deposition of atmospheric particles driven by the winter monsoon. Please see Lines 630-659, Page 21 for details.

  1. In conclusion, I suggest to describe the conclusion section much more concisely. Currently authors just listed the descriptions from the results.

      Response: The Conclusion has been improved to be more concise. In the revised Conclusion, we presented the major findings on (1) the performance of the combined method of three water body indices and a threshold and the JRC global surface water dataset for water body extraction, (2) the total area and trends of accretion and erosion and their influencing factors of the entire Tongtian River, and the reaches with the most erosion or accretion, (3) the performance of the lightweight neural network model. Please see Lines 725-744, Page 22-23 for details.

  1. Line 642, “, the source of Yangtze River,” could be deleted.

      Response: Revised as suggested. Please see Line 724, Page 22.

References:

  1. Chen, C.; Tian, B.; Schwarz, C.; Zhang, C.; Guo, L.; Xu, F.; Zhou, Y.; He, Q. Quantifying delta channel network changes with Landsat time-series data. Journal of Hydrology 2021, 600, 126688. [CrossRef]
  2. Hiraga, Y.; Kazama, S.; Ekkawatpanit, C.; Touge, Y. Impact of reclamation on the environment of the lower mekong river basin. Journal of Hydrology: Regional Studies 2018, 18, 143-155. [CrossRef]
  3. Beyaztas, U.; Shang, H.L.; Yaseen, Z.M. A functional autoregressive model based on exogenous hydrometeorological variables for river flow prediction. Journal of Hydrology 2021, 598, 126380. [CrossRef]
  4. Jamil, M.A.; Cabral, N.R.; Bezerra, S.Y.J.A.; Araújo, N.C.W.; de Sousa, M.W.; Germano, V.; Tales, T.; Bezerra, Y.J.A. Near-infrared spectroscopy for prediction of potentially toxic elements in soil and sediments from a semiarid and coastal humid tropical transitional river basin. Microchemical Journal 2022, 107544. [CrossRef]
  5. Huang, X.; Li, Y.; Tian, Z.; Ye, Q.; Ke, Q.; Fan, D.; Mao, G.; Chen, A.; Liu, J. Evaluation of short-term streamflow prediction methods in Urban river basins. Physics and Chemistry of the Earth, Parts A/B/C 2021, 123, 103027. [CrossRef]

Reviewer 2 Report

 Abstract

The paper only clarified the significance of the Tongtian River channel morphology, however, what is the main shortcoming of recent studies and what is the scientific contribution of this paper?

Please limit the number of abbreviations in the abstract, where the readers would be easily confused with too many abbreviations without the understanding of the paper. For example, “The erosion process of Tongtian River mainly occurred in Reaches A, B, D and E”, what did the alphabets mean?

Introduction

The objectives of this paper is not clear. Please clarify it briefly to show the main contribution of the paper.

Data

The serial Landsat images were used in this paper, including the Landsat 5,7 and 8. How to deal with the stripes of Landsat 7? The defect of this satellite would easily cause biases.

Method

How to combine four Landsat water body indices and the threshold (0.1) and the water extraction results of JRC for extracting the water information? This is important for the result and its accuracy is directly related to the final result. First, the author said that their calculation priority is MNDWI > EVI&MNDWI> 0.1 > EVI&NDVI. Secondly, Selecting different water body datasets for different river reaches. “reaches A, B, C, D adopts the water body datasets extracted using the four water body indices, and reaches E, F, G adopts the JRC-based water body datasets.” What is the logit and feasibility of this operation?

Result

Figures 5-11 show the dynamics of historic river erosion and siltation. Too many figures were used to show the one result, please condense the figures and the relevant content, where the main figure was shown.

Is there is accuracy assessment for the mapping of water information?

Conclusion

Condense the conclusion, which is too long at the present form. The conclusion is to show the most important findings of the paper for the readers.

Author Response

Reviewer #2

Comments and Suggestions for Authors

 Abstract

The paper only clarified the significance of the Tongtian River channel morphology, however, what is the main shortcoming of recent studies and what is the scientific contribution of this paper?

Please limit the number of abbreviations in the abstract, where the readers would be easily confused with too many abbreviations without the understanding of the paper. For example, “The erosion process of Tongtian River mainly occurred in Reaches A, B, D and E”, what did the alphabets mean?

Response: Thank you for this nice comment. We have revised the Abstract following this comment, but the shortcomings of recent studies are summarized in the Introduction rather than the abstract. In the revised Abstract we presented the key findings on the accretion and erosion area change and trend of the Tongtian River channel morphology, and the performance of the lightweight neural network. At the same time, we avoided the use of abbreviations in the Abstract. Please see Line 17-30, Page 1.

Introduction

The objectives of this paper is not clear. Please clarify it briefly to show the main contribution of the paper.

Response: Agree. We have rewritten the Introduction. Firstly, we reviewed the studies on channel morphology of the Tongtian River and found that previous studies seldom focused on (1) the erosion and accretion processes of the Tongtian River channel morphology at the large spatial scale and multi-temporal scales and (2) the prediction of the channel morphology of the Tongtian River. This study aims at bridging these two research gaps. Secondly, we reviewed the studies on different water body extraction methods and found that it is still unclear in previous studies whether the accuracy of the JRC global surface water dataset is better than the single water body index and the combined method of multiple water body indices. This is also a promising topic to be studied. Thirdly, a literature review was made for the investigation of the evolution of river channel morphology based on machine learning. We found that few studies have used the lightweight neural network (a simpler but more efficient method) to predict the accretion and erosion of the river channel morphology, which is another research objective of this study. This study intends to answer the following two key questions: (1) What are the spatio-temporal characteristics, trends and associated influencing factors of the accretion and erosion area change of the Tongtian River channel morphology? (2) Can the lightweight neural network model well predict the accretion and erosion area change of the Tongtian River channel morphology, and how is the prediction accuracy? Overall, the revised Introduction is much clearer and more logical on the literature review, research objectives, scientific questions, as well as the methodology. Please see Lines 37-158, Pages 1-4 for more details.

Data

The serial Landsat images were used in this paper, including the Landsat 5,7 and 8. How to deal with the stripes of Landsat 7? The defect of this satellite would easily cause biases.

Response: In order to reduce the impact of fringe noise in Landsat-7 images and improve the accu-racy of water body extraction from all remote sensing images, the water body images extracted from May to October each year through the combined method are synthesized using the reducer module in GEE to form an inter-annual water body image. The operation procedure is as follows: Sum up the pixel values in the same grid in all water images. When all pixels in the grid are summed, a composite image can be obtained. This part of description was added in the revised manuscript. Please see Lines 245-251, Page 7 for details.

Method

How to combine four Landsat water body indices and the threshold (0.1) and the water extraction results of JRC for extracting the water information? This is important for the result and its accuracy is directly related to the final result. First, the author said that their calculation priority is MNDWI > EVI&MNDWI> 0.1 > EVI&NDVI. Secondly, Selecting different water body datasets for different river reaches. “reaches A, B, C, D adopts the water body datasets extracted using the four water body indices, and reaches E, F, G adopts the JRC-based water body datasets.” What is the logit and feasibility of this operation?

Response: In the revised manuscript, we reviewed the studies on different water body extraction methods and found that it is still unclear in previous studies whether the accuracy of the JRC global surface water dataset is better than the single water body index and the combined method of multiple water body indices. This is a promising topic to be studied. Therefore, this study compared the different methods mentioned above and select the optimal method for water body extraction for different reach. The water body extraction methods are described in Sections 2.3 and 2.4. In Section 2.3, we mainly introduced the procedures of water body extraction using the combined methods of three water body indices and a threshold, and the JRC dataset. In the previous manuscript, we made a mistake on the number of water body indices used. We only used three indices (i.e., the MNDWI, NDVI and EVI), rather than four. We have corrected this mistake throughout the revised manuscript.

In Section 2.4, we compared the combined method, the JRC dataset, as well as the single water body index (the NDVI and EVI) for water body extraction of the seven reaches. We found that the combined method is the most accurate for water body extraction for the braided reaches compared to the JRC dataset and the single water body index such as the NDVI and EVI, while the JRC dataset outperform other methods for the meandering reaches. Further, we added one more sub-section 3.1 to assess the accuracy of the different water body extraction methods. The methodology is logical and feasible in this study.

Result

Figures 5-11 show the dynamics of historic river erosion and siltation. Too many figures were used to show the one result, please condense the figures and the relevant content, where the main figure was shown.

Is there an accuracy assessment for the mapping of water information?

Response: We have re-designed all the Figures 5-11 in the previous manuscript. Now, Figures 6-8 show the dynamics of historical river erosion and accretion at seven selected reaches during 1990-2020. The results for the three decadal episodes (1990-2000, 2000-2010, and 2010-2020) are shown in Figures S1-S7. The updated figures are of higher resolution and have clearer enlarged views. Please see the updated Figures 6-8 in the manuscript and the supplementary materials for details.

       In this study, we mainly assessed the accuracy of extracted water bodies qualitatively. The accuracy of extracted water body can be qualitatively assessed by visually comparing the extracted water body maps with the original Landsat images. The extracted water bodies with better connectivity and completeness are of higher accuracy. To better clarify the reliability of the methods used, we added one more sub-section 3.1 to compare the performance of different water body extraction methods for the braided reaches. Results indicate that the combined method is the most accurate for water body extraction for the braided reaches compared to the JRC dataset and the single water body index such as the NDVI and EVI, while the JRC dataset performs better than other methods for the meandering reaches. Considering the poor performance of the single water body index (i.e., the NDVI and EVI), the results of comparing the water bodies extracted for the meandering reaches using the JRC dataset and the single water body index were not presented in the manuscript.

Conclusion

Condense the conclusion, which is too long at the present form. The conclusion is to show the most important findings of the paper for the readers.

Response: The conclusion has been improved to be more concise. In the revised conclusion, we presented the major findings on (1) the performance of the combined method using three water bodies indices and a threshold and the JRC global surface water dataset for water body extraction, (2) the total area and trends of accretion and erosion and their influencing factors of the entire Tongtian River, and the reaches with the most erosion or accretion, (3) the performance of the lightweight neural network model. Please see Lines 725-744, Page 22-23 for details.

Reviewer 3 Report

Dear Authors, I really like your approach. The results sound plausible and have good discussion. After introducing the proposed changes, I believe that the work will be publishable.

Abstract: The summary should be corrected. Provide a general outline of the content of the work, including the essence of the studied problem, research methods and results. It is unacceptable as it stands. You give some vague specific markings like "Reaches A, B, D and E". GDF etc. Indicate where the research was conducted ......... Methods used and the new approach created .... Results obtained ..... How the proposed methodological approach can be used in other areas ....

Keywords: Limit to a few of the most important and fit-for-purpose results

Introduction: Lines 115-120 are only suitable for test methods Lines 126-132 are research methods

Study area and data Fig. 1. I do not know the Yangtze River, China! A demonstration of where your research area is situated in the background of the river basin.

The content of Tables 1 and 2 is not interesting. Is it necessary to post them?

Work structure: In Section 2.2 of Landsat images and JRC database, you discuss the research methods. You should change the structure of the article eg. 2. Material and Methods

2.1. Study area;

2.2. Landsat images ....;

2.3 Extracting water bodies ......

Author Response

Reviewer #3

Comments and Suggestions for Authors

Dear Authors, I really like your approach. The results sound plausible and have good discussion. After introducing the proposed changes, I believe that the work will be publishable.

Response: We thank the reviewer for the positive evaluation and insightful comments. We have improved the manuscript following your comments. Please see the point-by-point response below.

Abstract: The summary should be corrected. Provide a general outline of the content of the work, including the essence of the studied problem, research methods and results. It is unacceptable as it stands. You give some vague specific markings like "Reaches A, B, D and E". GDF etc. Indicate where the research was conducted ......... Methods used and the new approach created .... Results obtained ..... How the proposed methodological approach can be used in other areas ....

Response: Thank you for this nice comment. We have revised the Abstract following this comment. In the revised Abstract we presented the aim of this study and the key findings on the accretion and erosion area change and trend of the Tongtian River channel morphology, and the performance of the lightweight neural network. At the same time, we avoided the use of abbreviations and the vague specific markings like "Reaches A, B, D and E" in the Abstract. Please see Lines 17-30, Page 1 for more details.

Keywords: Limit to a few of the most important and fit-for-purpose results

Response: We have filtered and re-written the keywords. The five selected keywords are water body indices, lightweight neural network, river channel morphology, accretion and erosion area, the Tongtian River. Please see Lines 34-35, Page 1 for details.

Introduction: Lines 115-120 are only suitable for test methods Lines 126-132 are research methods

Response: We have rewritten the Introduction. Firstly, we reviewed the studies on channel morphology of the Tongtian River and found that previous studies seldom focused on (1) the erosion and accretion processes of the Tongtian River channel morphology at the large spatial scale and multi-temporal scales and (2) the prediction of the channel morphology of the Tongtian River. This study aims at bridging these two research gaps. Secondly, we reviewed the studies on different water body extraction methods and found that it is still unclear in previous studies whether the accuracy of the JRC global surface water dataset is better than the single water body index and the combined method of multiple water body indices. This is also a promising topic to be studied. Thirdly, a literature review was made for the investigation of the evolution of river channel morphology based on machine learning. We found that few studies have used the lightweight neural network (a simpler but more efficient method) to predict the accretion and erosion of the river channel morphology, which is another research objective of this study. This study intends to answer the following two key questions: (1) What are the spatio-temporal characteristics, trends and associated influencing factors of the accretion and erosion area change of the Tongtian River channel morphology? (2) Can the lightweight neural network model well predict the accretion and erosion area change of the Tongtian River channel morphology, and how is the prediction accuracy? Fourthly, we did not describe the detailed methods as done in the previous manuscript. Overall, the revised Introduction is much clearer and more logical on the literature review, research objectives, scientific questions, as well as the methodology. Please see Lines 37-158, Pages 1-4 for more details.

Study area and data Fig. 1. I do not know the Yangtze River, China! A demonstration of where your research area is situated in the background of the river basin.

Response: The Yangtze River is the largest river in Asia and China, and the third largest river in the world. We have updated the Figure 1 by inserting a location map for the Tongtian River. Also, the major rivers of the Yangtze River are also shown in the location map. Please see Line 210, Page 5 for details.

The content of Tables 1 and 2 is not interesting. Is it necessary to post them?

Response: We have moved those two tables into the supplementary materials.

Work structure: In Section 2.2 of Landsat images and JRC database, you discuss the research methods. You should change the structure of the article eg. 2. Material and Methods

2.1. Study area;

2.2. Landsat images ....;

2.3 Extracting water bodies ......

      Response: We have changed the structure of the article following this comment. The revised manuscript has five major sections and seven sub-sections for the Materials and Methods.

Round 2

Reviewer 1 Report

Authors substantially revised the manuscript addressing the comments and the revised version of manuscript is much improved. However, I still have some minor comments that need to be addressed.

(1)  In equation 3, please make “Blue” into italic form, and also in line 244, please make Green, Blue, NIR, R, MIR into italics form as all the variables used in equation should be written in italics.

(2)  Lines 293-294, I think, the values 4.4 or 1.71 degree of water body width of the braided reach is too large. Please check and describe it clearly. Also, please clarify why the water body width of the braided reach is expressed in degree not in meter? 

(3)  Figure 3, please remove white filled color in all smaller red rectangles, make it transparent.

(4)  Line 311, shp – shape?

(5)  Line 425, please delete “accurate and”. How accurate is it?

Author Response

Dear Reviewer,

We have revised the manuscript following your comments. In addition, we have carefully checked the presentation throughout the manuscript. Some grammar mistakes and other language problems have been corrected. We also re-organized the sections 2.4 and 2.5 to make the structure more logical.

Please kindly find the point-by-point response below. Your nice comments have helped us improve the manuscript quite a lot. Hope the revised manuscript can meet the quality requirement of Remote Sensing. Please feel free to share us your further comments and advices to improve the paper, if any. Thanks for your time and contribution.

Best regards,

Bin Deng, Kai Xiong, Zhiyong Huang,

Changbo Jiang, Jiang Liu, Wei Luo, and Yifei Xiang

Reviewer #1

Comments and Suggestions for Authors

General Comments:

Authors substantially revised the manuscript addressing the comments and the revised version of manuscript is much improved. However, I still have some minor comments that need to be addressed.

Specific Comments:

  • In equation 3, please make “Blue” into italic form, and also in line 244, please make Green, Blue, NIR, R, MIR into italics form as all the variables used in equation should be written in italics.

Response: Agree. Revised as suggested. All the variables used in equation were revised to be italics. Please see Line 247, Page 7 for details.

  • Lines 293-294, I think, the values 4.4 or 1.71 degree of water body width of the braided reach is too large. Please check and describe it clearly. Also, please clarify why the water body width of the braided reach is expressed in degree not in meter? 

Response: We thank the reviewer for this insightful comment. After a careful check, we found some mistakes on the calculation of the water body width in the previous manuscript. To avoid controversy and misinterpretation, we have deleted this part of information. The corresponding sentences that followed were revised to be more logical. Please see Lines 308-317, Page 8.

  • Figure 3, please remove white filled color in all smaller red rectangles, make it transparent.

Response: Revised as suggested. There was a problem with the format of Figure 3 in the previous manuscript, leading to an error when transforming the word to pdf. We have corrected this problem in the revised manuscript. Please see Line 339, Page 9.

(4)  Line 311, shp – shape?

Response: Revised as suggested. Please see Line 336, Page 9.

(5) Line 425, please delete “accurate and”. How accurate is it?

Response: We have deleted “accurate and”. The word “accurate” has a quantitative meaning, which is difficult to be achieved due to the lack of in situ measurements of the water bodies. In this study, the accuracy assessment was perform qualitatively by visual interpretation of Landsat images. Therefore, it is better to just say “reliable” rather than “accurate” as suggested by this comment. Please see Line 457, Page 12.

This manuscript is a resubmission of an earlier submission. The following is a list of the peer review reports and author responses from that submission.